# Stable centromere association of the yeast histone variant Cse4 requires its essential N-terminal domain

Andrew R Popchock[1], Sabrine Hedouin[1], Yizi Mao[2], Charles L Asbury [ID][3], Andrew B Stergachis[2,4] & Sue Biggins [ID][1✉]

## Abstract

**Chromosome segregation relies on kinetochores that assemble on specialized centromeric chromatin containing a histone H3 variant. In budding yeast, a single centromeric nucleosome containing Cse4 assembles at a sequence-defined 125 bp centromere. Yeast centromeric sequences are poor templates for nucleosome formation in vitro, suggesting the existence of mechanisms that specifically stabilize Cse4 nucleosomes in vivo. The extended Cse4 N-terminal tail binds to the chaperone Scm3, and a short essential region called END within the N-terminal tail binds the inner kinetochore complex Okp1/Ame1. To address the roles of these interactions, we utilized single-molecule fluorescence assays to monitor Cse4 during kinetochore assembly. We found that Okp1/Ame1 and Scm3 independently stabilize Cse4 at centromeres via their END interaction. Scm3 and Cse4 stability at the centromere are enhanced by Ipl1/Aurora B phosphorylation of the Cse4 END, identifying a previously unknown role for Ipl1 in ensuring Cse4 stability. Strikingly, a phosphomimetic mutation in the Cse4 END restores Cse4 recruitment in mutants defective in Okp1/Ame1 binding. Together, these data suggest that a key function of the essential Cse4 N-terminus is to ensure Cse4 localization at centromeres.**

**Keywords** Centromeric Nucleosome; Centromere; Kinetochore; Chromosome Segregation; TIRF Microscopy
**Subject Categories** Cell Cycle; Chromatin, Transcription & Genomics

## Introduction

During cell division, chromosomes are replicated to form sister chromatids and then segregated into daughter cells. Chromosome segregation harnesses the forces generated by spindle microtubules through the kinetochore, a conserved megadalton protein machine. Kinetochores contain dozens of multiprotein subunits that assemble on centromeric chromatin containing a specialized histone H3 variant called CENP-A (Palmer et al, 1991; Régnier

et al, 2005; Sullivan et al, 1994). Remarkably, kinetochores must assemble de novo at centromeres every cell cycle after replication (Müller and Almouzni, 2014; Pan et al, 2019; Wisniewski et al, 2014). Most eukaryotes have large regional centromeric regions containing interspersed CENP-A and canonical H3 nucleosomes that serve as the platform for the recruitment of chromosome-adjacent inner kinetochore proteins to form the constitutive centromere associated network (CCAN) (Barnhart et al, 2011; Blower et al, 2002; Foltz et al, 2006; Weir et al, 2016; Yatskevich et al, 2022), which then facilitates assembly of the outer kinetochore that interacts with spindle microtubules (Dhatchinamoorthy et al, 2017; Polley et al, 2024). In contrast, budding yeast have a unique point centromere that is sequence-defined and recruits a single CENP-A nucleosome that mediates the assembly of the entire kinetochore (Bloom and Carbon, 1982; Clarke and Carbon, 1980; Fitzgerald-Hayes et al, 1982; Furuyama and Biggins, 2007; Joglekar et al, 2006; Meluh et al, 1998). Because yeast kinetochores assemble on a short, defined sequence with a single nucleosome, they are an ideal model system to study kinetochores since key aspects of their architecture and function are conserved.

Although a single yeast centromeric nucleosome is sufficient for kinetochore assembly, yeast centromeric DNA is a poor template for the reconstitution of CENP-A (Cse4) nucleosomes using purified components in vitro (Drew and Travers, 1985; Meluh et al, 1998; Mizuguchi et al, 2007; Widom, 2001). Reconstitution of yeast centromeric nucleosomes requires additional stabilizing factors to form a stable nucleosome-core particle (Guan et al, 2021), or alterations of the native sequence in larger kinetochore reconstitutions (Dendooven et al, 2023; Yan et al, 2019). The inherent unfavourability for stable centromeric nucleosome formation is counter to their critical function of kinetochore assembly in cells but has been proposed as a possible regulatory function to exclude canonical nucleosomes (Dechassa et al, 2014). Because centromere nucleosome assembly must occur rapidly and with high fidelity after replication, factors must exist that promote this process in vivo. Consistent with this requirement, we recently found that Cse4 stabilization requires two additional kinetochore proteins, the conserved histone chaperone Scm3 (HJURP) and the essential kinetochore complex Okp1/Ame1 (CENP-QU) (Popchock et al, 2023), although the mechanisms by which they stabilize the centromeric nucleosome are not known.

[1]Howard Hughes Medical Institute, Basic Sciences Division, Fred Hutchinson Cancer Center, Seattle, WA 98109, USA. [2]Division of Medical Genetics, Department of Medicine, University of Washington, Seattle, WA, USA. [3]Department of Physiology and Biophysics, University of Washington, Seattle, WA, USA. [4]Department of Genome Sciences, University of Washington, Seattle, WA, USA. ✉E-mail: sbiggins@fredhutch.org

    

Scm3 and the Okp1/Ame1 complex directly interact with Cse4. The Scm3 chaperone was initially found to bind to a Cse4 region in the histone fold domain-containing residues 166–201 called the centromere-associated targeting domain (CATD) (Cho and Harrison, 2011; Dechassa et al, 2011; Xiao et al, 2011; Zhou et al, 2011). However, a recent study reported that Scm3 also binds to Cse4's highly disordered extended N-terminal domain (NTD, Cse4-1-133) which contains the initial 133 residues of Cse4 (Shukla et al, 2024). The NTD contains a short essential region spanning residues 28–60, named the essential N-terminal domain (END, Cse4-28-60) (Chen et al, 2000) and it is the target of several different types of post-translational modifications (Anedchenko et al, 2019; Boeckmann et al, 2013; Mishra et al, 2021), but the functional consequences of these modifications remain largely unknown. The END contains an essential Okp1/Ame1 binding interface (Anedchenko et al, 2019; Chen et al, 2000; Deng et al, 2023; Fischböck-Halwachs et al, 2019; Hinshaw and Harrison, 2019) and was also shown to undergo significant structural rearrangement in reconstitutions containing Cse4-NTD bound to Scm3 (Shukla et al, 2024). The additional Scm3 interaction site in Cse4 expands the potential functional roles of Scm3 at the kinetochore beyond its canonical role in Cse4 centromere targeting through the conserved CATD domain. This additional function may depend upon its interaction with the N-terminal tail of Cse4, a region which has been shown to play a role in the localization of CENP-A to centromeres in *C. elegans* (de Groot et al, 2021; Prosee et al, 2021). This is consistent with our recent findings that Scm3 can stabilize the interaction of Cse4 with centromeric DNA after initial association during de novo kinetochore assembly by an unknown mechanism (Popchock et al, 2023). An additional function for Scm3 at centromeres may also explain its behavior in cells, where it has been shown to be in constant exchange at kinetochores throughout mitosis, long after centromeric targeting and stable Cse4 incorporation (Wisniewski et al, 2014).

To elucidate the function of Scm3 and Okp1/Ame1 binding to the essential N-terminus of Cse4, we utilized our recently developed single-molecule fluorescence assay that assembles centromeric nucleosomes in yeast lysates. In contrast to conventional biochemical reconstitutions that do not achieve stable centromeric nucleosomes without alterations or stabilizing techniques, Cse4 is rapidly assembled onto centromeric DNA sequences in yeast extract and these native nucleosomes are remarkably stable when removed from extract (Popchock et al, 2023). Here we show that the END region is required for stable Cse4 association with centromeric DNA and requires the independent recruitment of both Okp1/Ame1 and Scm3. In cells, we find that disruption of END binding to Okp1/Ame1 causes a reduction of Cse4 centromeric levels that can be partially rescued through Ipl1-mediated phosphorylation of the END domain. Taken together, our data indicate that the Cse4 END not only recruits proteins to assemble the kinetochore, but that END recruitment of these proteins stabilizes the centromeric nucleosome. We propose that this multistep stabilization mechanism not only ensures that cells assemble a single Cse4 nucleosome at centromeres, but also makes it difficult to stabilize ectopic centromeric nucleosomes that could lead to genomic instability.

# Results

## Okp1/Ame1 contributes to Cse4 stability in vivo

To test if the recruitment of Okp1/Ame1 contributes to Cse4 stability in cells, we utilized a technique called Fiber-seq that enables nucleosome mapping at the single-molecule level on chromatin fibers (Jha et al, 2024; Stergachis et al, 2020) (Fig. 1A). Briefly, spheroplasted cells were permeabilized and treated with a nonspecific N6-adenine DNA methyltransferase (m6A-MTase) that methylates accessible (non-protein bound) DNA. Individual m6A-stenciled chromatin fibers are sequenced using long-read single-molecule sequencing to >100x genomic coverage to identify single-molecule protein occupancy events at single-molecule and near single-nucleotide resolution. The entirety of each point centromere in budding yeast is readily captured along a sequenced read, which are >10 kb in length, and as this is a genome-wide method, each of the 16 centromeres was thoroughly captured by sequenced fibers (~1100 fibers for WT cells and ~500 fibers for mutants). Consequently, this approach enabled the direct quantification of the average steady-state absolute occupancy of the centromeric nucleosome across individual fibers in yeast cells (Fig. 1B). We began by testing whether differences in Cse4 occupancy levels at centromeres could be detected under the most stringent conditions, when Cse4 is depleted using a *cse4-AID* system. We performed Fiber-seq on WT and Cse4-depleted cells and found nearly full nucleosome occupancy (92.2%) at centromeres in WT cells, which was markedly reduced to 41.6% in *cse4-AID* cells (Fig. 1B). We then depleted Okp1/Ame1 in cells using an *okp1-AID* strain to determine if Cse4 nucleosome occupancy was affected. After Okp1/Ame1 depletion, nucleosome occupancy levels at centromeres were significantly reduced (56.0%) (Fig. 1B). This indicated a loss of Cse4 at centromeres after Okp1/Ame1 depletion that was consistent with our previous observations in de novo reconstitutions (Popchock et al, 2023), suggesting a role for Okp1/Ame1 in Cse4 retention at centromeres.

## The Cse4 END is critical for Cse4 recruitment to centromeric DNA

Because Okp1/Ame1 is required for stable Cse4 association with centromeric DNA in vivo, we asked whether this is mediated by its essential interaction with the END (Deng et al, 2023). To test this, we first asked whether the END contributes to Cse4 stability at the centromere. We monitored Cse4 centromeric recruitment using a recently developed single-molecule TIRFM de novo kinetochore assembly assay (Popchock et al, 2023) (Fig. 2A). Briefly, fluorescently labeled centromeric DNA templates (CEN DNA) are sparsely attached to a coverslip surface and then cell extract containing endogenously tagged fluorescent kinetochore protein(s) is introduced into a flow chamber and incubated for 90 min. After incubation to allow kinetochore assembly, the extract is washed away and the colocalization of the target fluorescent protein with the labeled CEN DNA is quantified. We introduced an ectopic copy of GFP-tagged wild-type Cse4 (*pGAL-CSE4-GFP)*, or a deletion mutant lacking the END, Cse4$^{\Delta END}$ (*pGAL-cse4$^{\Delta END}$-GFP*) (Fig. 2B), under a galactose inducible promoter to overexpress them during a mitotic arrest. We introduced the GFP tag into an internal site within the Cse4 N-terminus to retain functionality (Wisniewski et al, 2014). Ectopic expression was needed because the Cse4 END is essential for viability. The proteins were overexpressed relative to the endogenous Cse4 protein (Fig. EV1A). After incubation with *pGAL-CSE4-GFP* extract for 90', ~35% of the CEN DNA exhibited colocalization with overexpressed Cse4-GFP. In contrast, the endpoint colocalization of overexpressed Cse4$^{\Delta END}$-GFP was

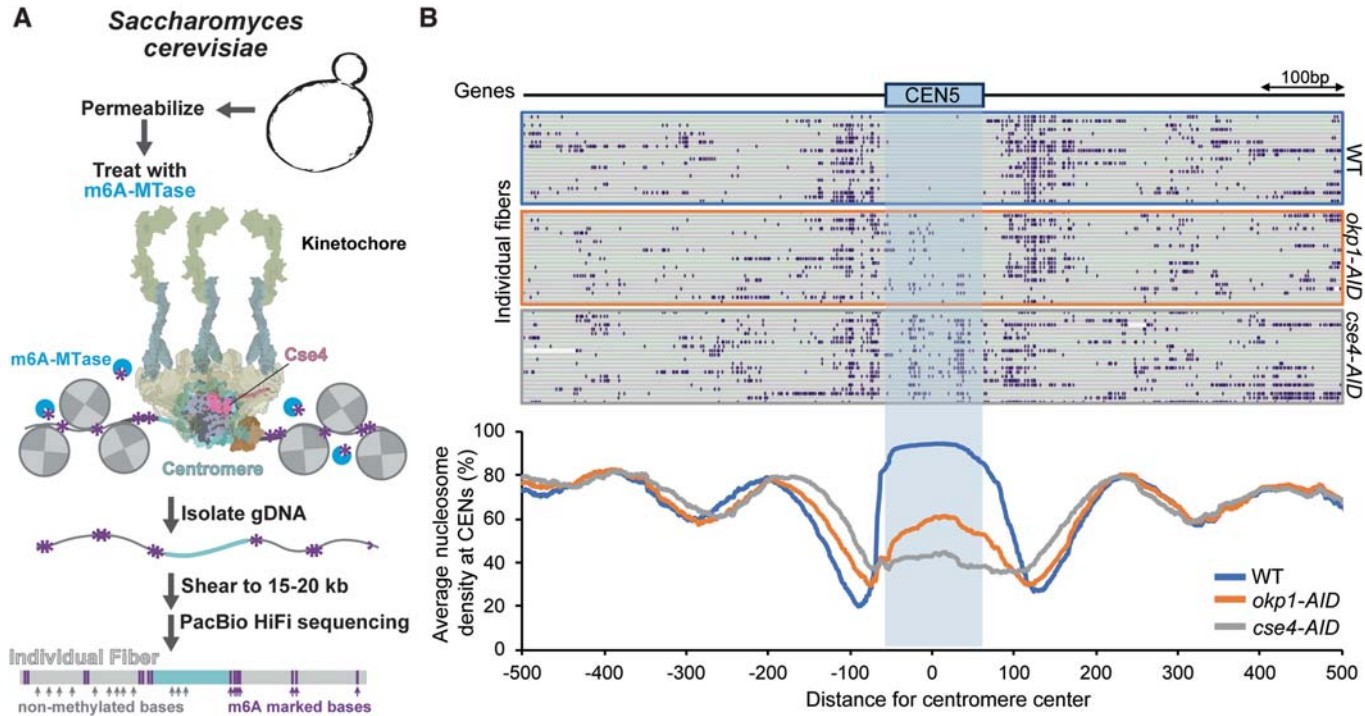

**Figure 1. Depletion of Okp1 significantly reduces Cse4 nucleosome density at centromeres in cells.**

(A) Schematic representing experimental design for Fiber-seq of *Saccharomyces cerevisiae* including example centromeric locus with a kinetochore-bound centromeric nucleosome. (B) Example Fiber-seq reads of chromatin fibers with methylated bases shown in purple through centromere regions of chromosome V in WT (SBY3, top), *okp1-AID* (SBY15124, middle), and *cse4-AID* (SBY22656, bottom) cells. Bottom graph indicates the calculated average nucleosome density for all chromosomes centered at centromeres from Fiber-seq genomic DNA analysis from *CSE4-GFP* (blue), *okp1-AID* (orange), and *cse4-AID* (gray) cells.

significantly reduced to 12% (Fig. 2B) despite similar propensities to colocalize with CEN DNA during incubation (Fig. EV1B), indicating a role for the END in stable Cse4 centromere recruitment. Because prior structural studies with reconstituted nucleosomes do not provide a mechanism for how the END would stabilize Cse4 and do not use native centromere sequences (Dendooven et al, 2023; Yan et al, 2019), we asked whether the native kinetochore assembly pathway differs from what occurs during conventional reconstitutions. To do this, we tested the non-native centromere sequence used in the most recent structural studies (C0N3 DNA) in TIRFM kinetochore assembly assays (Fig. 2C) (Dendooven et al, 2023). We found that despite slightly enhanced recruitment of the CBF3 component Ndc10 (Fig. EV1C), Cse4 levels were significantly lower on C0N3 DNA when compared to the native centromere sequence (Fig. 2C), suggesting that the assembly pathway of reconstituted kinetochores using purified proteins may differ from those containing native components. This significant drop in Cse4 levels observed on C0N3 DNA is much greater than the variance we would expect between native centromere sequences based on previous observations and is comparable to mutant centromere sequences that are unstable in cells (Popchock et al, 2023). In addition, we did not detect any competition from H3 that could explain the low Cse4 association when we analyzed a similar template (Popchock et al, 2023). Together, these data identify a critical role for the Cse4 END domain in its stable centromere association, a finding that was not previously apparent from structural reconstitutions.

## Okp1/Ame1 binding to Cse4 END stabilizes the centromeric nucleosome

To determine whether Cse4 stabilization by the END domain is due to its interaction with Okp1/Ame1, we turned to recently reported Cse4 mutations at L41 that disrupt Okp1/Ame1 binding (Deng et al, 2023; Hinshaw and Harrison, 2019). Consistent with this data, we found that Cse4-NTD$^{L41A}$ and Cse4-NTD$^{L41D}$ disrupted Okp1/Ame1 binding in pull-down assays (Fig. 3A). Although the Cse4-L41D mutant more strongly disrupted Okp1/Ame1 binding, it was not used for further study because it is inviable (Deng et al, 2023) and the use of endogenous Cse4 alleles was necessary for monitoring Cse4 behavior with respect to other kinetochore proteins. We next analyzed Okp1/Ame1 localization to centromeres by performing bulk kinetochore assembly assays, where CEN DNA templates are linked to magnetic beads and directly incubated in cell extracts (Lang et al, 2018). We found that cell extracts containing the Cse4$^{L41A}$ mutant completely abolished Okp1/Ame1 complex recruitment (Fig. EV2A), indicating that Okp1/Ame1 kinetochore assembly requires binding to the Cse4 END. Consistent with an additional role for Okp1/Ame1 in stabilizing Cse4, we found that Cse4 centromere recruitment was also significantly reduced in Cse4$^{L41A}$ mutant extracts (Fig. EV2A). However, the recruitment of Ndc10, which is necessary for Cse4 deposition, was not affected (Fig. EV2A). To quantify the Cse4 recruitment defect, we performed TIRFM endpoint colocalization assays. Consistent with the bulk assembly assays, Cse4$^{L41A}$-GFP endpoint

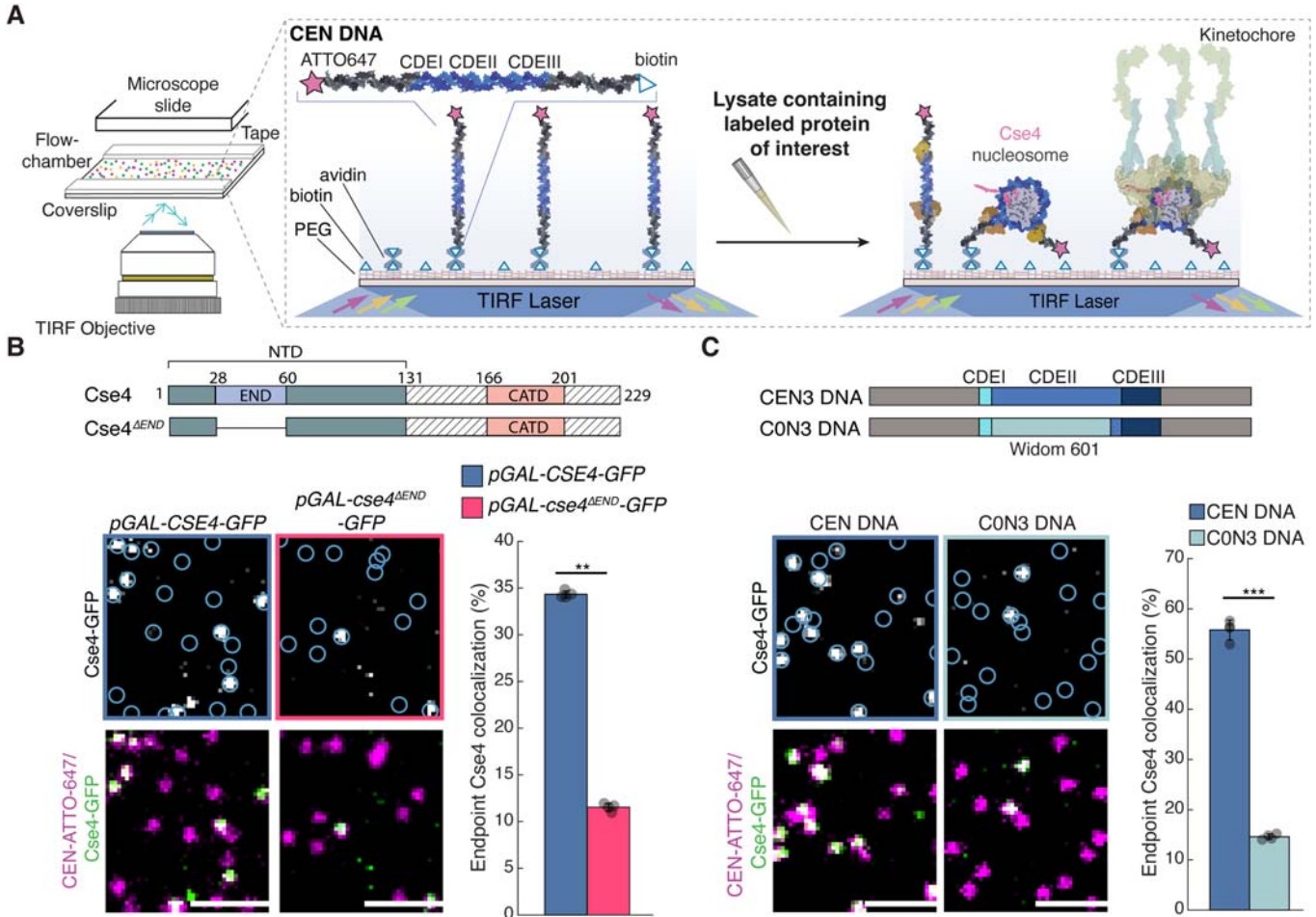

**Figure 2. Cse4 centromeric localization depends on both the Cse4 END region and native centromeric sequence.**

(A) Schematic diagram of de novo TIRFm kinetochore assembly assays (adapted from (Popchock et al, 2023)). (B) Cse4 construct schematic (top) and example images of TIRFM endpoint colocalization assays. Top panels show visualized Cse4-GFP on CEN DNA in *pGAL-CSE4-GFP* (SBY22273) extracts (top left) or *pGAL-cse4^{ΔEND}-GFP* (SBY22803) extracts (top right) with colocalization shown in relation to identified CEN DNA in blue circles. Bottom panels show respective overlays of CEN DNA channel (magenta) with Cse4-GFP (green). Scale bars 2 μm. Graph indicates Cse4-GFP endpoint colocalization with CEN DNA in extracts from *pGAL-CSE4-GFP* or *pGAL-cse4-ΔEND-GFP* genetic backgrounds (34 ± 0.4%, 12 ± 0.4%, avg ± s.d. n = 4 experiments, each examining ~1000 DNA molecules from different extracts, ** indicates significant difference with two-tailed P value of 1.0E-3). (C) DNA template schematic (top) and example images of TIRFM endpoint colocalization assays. Top panels show visualized Cse4-GFP on CEN DNA (top-left panel), or on C0N3 DNA (top-right panel) in *CSE4-GFP* (SBY21863) extracts with colocalization shown in relation to identified DNAs in blue circles. Bottom panels show overlay of CEN or C0N3 DNA channel (magenta) with Cse4-GFP (green), Scale bars 2 μm. Graph indicates quantification of Cse4-GFP endpoint colocalization with CEN DNA or C0N3 DNA (56 ± 2.0%, 15 ± 0.5%, avg ± s.d. n = 4 experiments, each examining ~1000 DNA molecules from different extracts, *** indicates significant difference with two-tailed P value of 3.5E-5).

colocalization was significantly lower than Cse4-GFP (7% vs 55%, respectively, Fig. 3B). To confirm this was due to disruption of the binding interface between END and Okp1/Ame1 (Fig. EV2B), we introduced an orthogonal mutation in the Okp1/Ame1 complex component protein Ame1 at residue I195 (*ame1^{I195Y}*), which was previously shown to abrogate Okp1/Ame1 binding to the Cse4 END at the same interface as the Cse4-L41 mutants via disruption in hydrophobic packing (Deng et al, 2023). In endpoint colocalization assays, Cse4-GFP levels were significantly reduced in extracts containing the Ame1^{I195Y} mutant relative to WT (17% vs 55%, respectively, Fig. 3B). To confirm that Cse4 recruitment was dependent upon this unique binding interface between END and Okp1/Ame1, we also monitored Cse4 colocalization in extracts lacking the CCAN component Ctf19 (Δ*ctf19*) and found no

significant disruption in Cse4 colocalization levels (Fig. EV2C). Together, these results indicate that END recruitment of Okp1/Ame1 is needed for stable Cse4 association with centromeric DNA.

To further explore Cse4 stabilization by Okp1/Ame1 binding, we sought to dissect the behavior of Cse4 on centromeric DNA in relation to Okp1/Ame1. To do this, we utilized our previously developed time-lapse assay that enables direct and simultaneous monitoring of two orthogonally labeled fluorescent kinetochore proteins in cell extract with CEN DNA over time (Popchock et al, 2023). Continuous monitoring allows us to analyze more complex and dynamic binding behavior based on simultaneous protein–protein colocalization. We therefore orthogonally labeled Ame1-mKate2 and Cse4-GFP and simultaneously monitored their behavior to identify instances when they formed a ternary complex

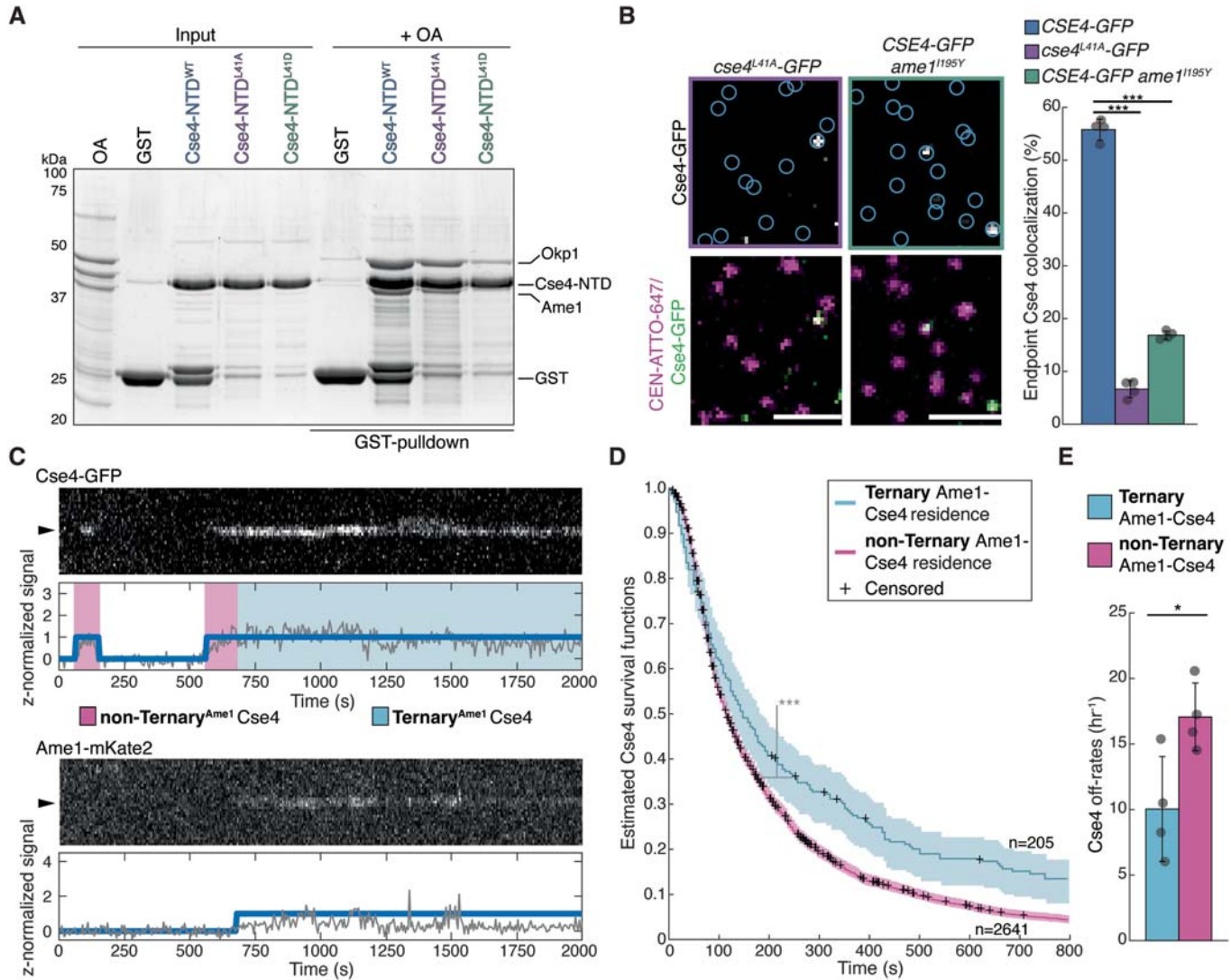

**Figure 3. Okp1/Ame1 recruitment by Cse4 END is required for stable Cse4 localization to centromeric DNA.**

(A) SDS-PAGE of GST pull-down assays of immobilized Cse4-NTD$^{WT}$, Cse4-NTD$^{L41A}$ and Cse4-NTD$^{L41D}$ to test binding of recombinant Okp1/Ame1. (B) Example images of TIRFM endpoint colocalization assays. Top panels show visualized Cse4-GFP on CEN DNA in *cse4$^{L41A}$-GFP* (SBY22811) extracts (top-left panel) or *CSE4-GFP ame1$^{I195Y}$* (SBY23105) extracts (top-right panel) with colocalization shown in relation to identified CEN DNA in blue circles. Bottom panels show overlay of CEN DNA channel (magenta) with Cse4-GFP (green), scale bars 2 µm. Graph indicates quantification of Cse4-GFP endpoint colocalization with CEN DNA in extracts from *CSE4-GFP*, *cse4$^{L41A}$-GFP* or *CSE4-GFP ame1$^{I195Y}$* genetic backgrounds (56 ± 2.0%, 6.6 ± 1.6% and 16.8 ± 0.8% avg ± s.d. *n* = 4 experiments, each examining ~1000 DNA molecules from different extracts, *** indicates significant difference with two-tailed *P* value between *CSE4-GFP* and *cse4$^{L41A}$-GFP* of 2.1E-8 and between *CSE4-GFP* and *CSE4-GFP ame1$^{I195Y}$* of 3.5E-6). (C) Representative residence lifetime assay trace of Cse4-GFP and Ame1-mKate2 on a single CEN DNA in *CSE4-GFP AME1-mKate2 extract* (SBY22244). Top panel includes kymograph of Cse4 (top–488 nm) in relation to single identified CEN DNA (arrow), with normalized intensity trace (gray-bottom) as well as identified residences (blue). Bottom panel includes kymograph of Ame1 (bottom—561 nm) in relation to the same identified CEN DNA (arrow), with normalized intensity trace (gray-bottom) as well as identified residences (blue). Images acquired every 5 s with normalized fluorescence intensity shown in arbitrary units. (D) Estimated survival function plots of Kaplan–Meier analysis of the lifetimes of Ternary Ame1-Cse4 residences on CEN DNA (blue—median lifetime of 147 s, *n* = 205 over four experiments of ~1000 DNA molecules using different extracts) and non-Ternary Ame1-Cse4 residences on CEN DNA (purple—median lifetime of 113 s, *n* = 2641 over four experiments of ~1000 DNA molecules using different extracts). 95% confidence intervals indicated (shaded lines), right-censored lifetimes (plus icons) were included with equivalent weighting in survival function estimates, *** indicates a significant difference two-tailed *P* value of 3.0e-06 as determined by log-rank test. (E) Graph indicates quantification of the estimated off-rates of Ternary Ame1-Cse4 and non-Ternary Ame1-Cse4 residences on CEN DNA (10.0 h$^{-1}$± 4.0 h$^{-1}$ and 17.1 ± 2.6 h$^{-1}$ respectively, avg ± s.d. *n* = 3598 over four experiments of ~1000 DNA molecules, * indicates significant difference with two-tailed *P* value of 0.03).

on centromeric DNA (Fig. 3C). We then compared the residence times of Cse4 on CEN DNA in the presence of Okp1/Ame1 (Ternary Ame1-Cse4) or absence of Okp1/Ame1 (non-Ternary Ame1-Cse4) (Fig. 3C). Kaplan–Meier analysis revealed a significant increase in the median lifetime of Ternary Ame1-Cse4 versus non-

Ternary Ame1-Cse4 (147 s vs. 113 s, respectively, Fig. 3D). Consistent with this, the off-rates of Ternary Ame1-Cse4 were significantly slower (indicating more stable association) on centromeric DNA when compared to non-Ternary Ame1-Cse4 (10.0 h$^{-1}$ vs 17.1 h$^{-1}$, Fig. 3E). These data indicate that the

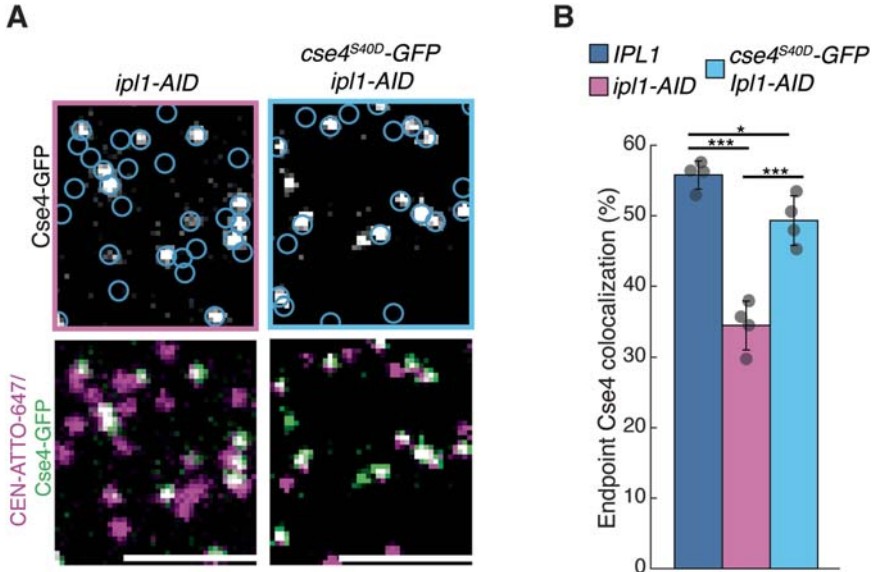

**Figure 4.  Ipl1 contributes to Cse4 localization via END phosphorylation.**

(A) Example images of TIRFM endpoint colocalization assays. Top panels show visualized Cse4-GFP on CEN DNA in *CSE4-GFP ipl1-AID* (SBY21971) extracts (top left) and *cse4^{S40D}-GFP ipl1-AID* (SBY20021) extracts (top right) with colocalization shown in relation to identified CEN DNA in blue circles. Bottom panels show overlay of CEN DNA channel (magenta) with Cse4-GFP (green), Scale bars 3 µm. (B) Graph indicates quantification of Cse4-GFP endpoint colocalization with CEN DNA in *IPL1* (SBY21863), *CSE4-GFP ipl1-AID* extracts and *cse4^{S40D}-GFP ipl1-AID* extracts (56 ± 2.0%, 34 ± 3.5% and 49 ± 3.5%, avg ± s.d. $n = 4$ experiments, each examining ~ 1,000 DNA molecules from different extracts, *** indicates significant difference with two-tailed *P* value between *IPL1* and *CSE4-GFP ipl1-AID* of 1.3E-4 and between *cse4^{S40D}-GFP ipl1-AID* and *CSE4-GFP ipl1-AID* of 9.7E-4, * indicates significant difference between *IPl1* and *cse4^{S40D}-GFP ipl1-AID* with two-tailed *P* value of 0.02).

formation of a complex between Cse4 and Okp1/Ame1 stabilizes Cse4 on CEN DNA, providing an explanation for the reduced endpoint localization of Cse4^{L41A} to CEN DNA.

## The conserved mitotic kinase Ipl1 contributes to stable Cse4 centromeric localization via END domain phosphorylation

The END domain is phosphorylated by the Polo kinase (Cdc5) on S33 and the Aurora B (Ipl1) kinase on Cse4-S40 (Boeckmann et al, 2013; Mishra et al, 2019), so we next asked whether Cdc5 and Ipl1 regulate Cse4 centromeric localization. We monitored Cse4 colocalization levels using the TIRFM kinetochore assembly assays in cell extracts that were depleted of Cdc5 (*cdc5-AID*) and found no significant change in Cse4 colocalization levels (Appendix Fig. S1). In contrast, when extracts had been depleted of Ipl1 (*ipl1-AID*), we found Cse4 recruitment to centromeric DNA was reduced (Fig. 4A,B). To test whether the Ipl1-dependent requirement was dependent upon Cse4 END phosphorylation, we asked if a phosphomimetic mutant could restore Cse4 localization in the absence of Ipl1. Strikingly, Cse4^{S40D} partially restored Cse4 levels in Ipl1-depleted extracts in the TIRFM assembly (Fig. 4A,B), suggesting that Ipl1 may act at least in part through END domain phosphorylation.

## Scm3 binds to the END domain and Ipl1 phosphorylation enhances the interaction

Although the Cse4 END domain binds to Okp1/Ame1, phosphorylation of Cse4 at S40 is not predicted to affect the interaction based on structural studies (Fig. EV2B) and we did not detect a change in binding a pull-

down assay (Fig. 5A). We also performed TIRFM endpoint colocalization assays to measure Okp1/Ame1 recruitment (using Ame1-GFP) in extracts containing either Cse4 or the phosphomimetic Cse4^{S40D} mutant. Consistent with the pull-down assays, Ame1-GFP colocalization levels were similar in extracts containing the Cse4^{S40D} mutant when compared to WT Cse4 (Appendix Fig. S2A). It was recently reported that the Cse4 N-terminus also binds to the Scm3 chaperone (Shukla et al, 2024), so we tested whether the END domain is required for Scm3 association in a pull-down assay. We generated recombinant GST-fusions to the entire Cse4 N-terminal domain (Cse4-NTD^{WT}) as well as the NTD lacking the END region (Cse4-NTD^{ΔEND}) to use as bait in pull-down assays (Fig. 5B). We first confirmed that recombinant Okp1/Ame1 binding to the Cse4-NTD^{WT} was completely lost in the Cse4-NTD^{ΔEND} mutant, consistent with previous studies (Fig. 5B). We then performed pull-down assays with recombinant Scm3 and found that it readily bound Cse4-NTD^{WT}, but its binding was significantly disrupted in the Cse4-NTD^{ΔEND} mutant (Fig. 5B). We therefore tested if Scm3:END binding was affected by Ipl1 phosphorylation in pull-down assays. To do this, we tested if the phosphomimetic Cse4-NTD^{S40D} enhanced Scm3:END binding in pull-down assays. Scm3 binding to Cse4-NTD^{S40D} was stimulated relative to Cse4-NTD under varying concentrations of purified recombinant Scm3 (Appendix Fig. S2B), suggesting that phosphorylation of Cse4 may regulate Scm3 binding to the END.

## Phosphorylation of the END domain promotes stable Cse4 localization

We next asked if phosphorylation of the Cse4 END stabilizes Cse4 on centromeres. First, we monitored the localization of phosphomimetic Cse4^{S40D} and phosphonull Cse4^{S40A} mutants in bulk

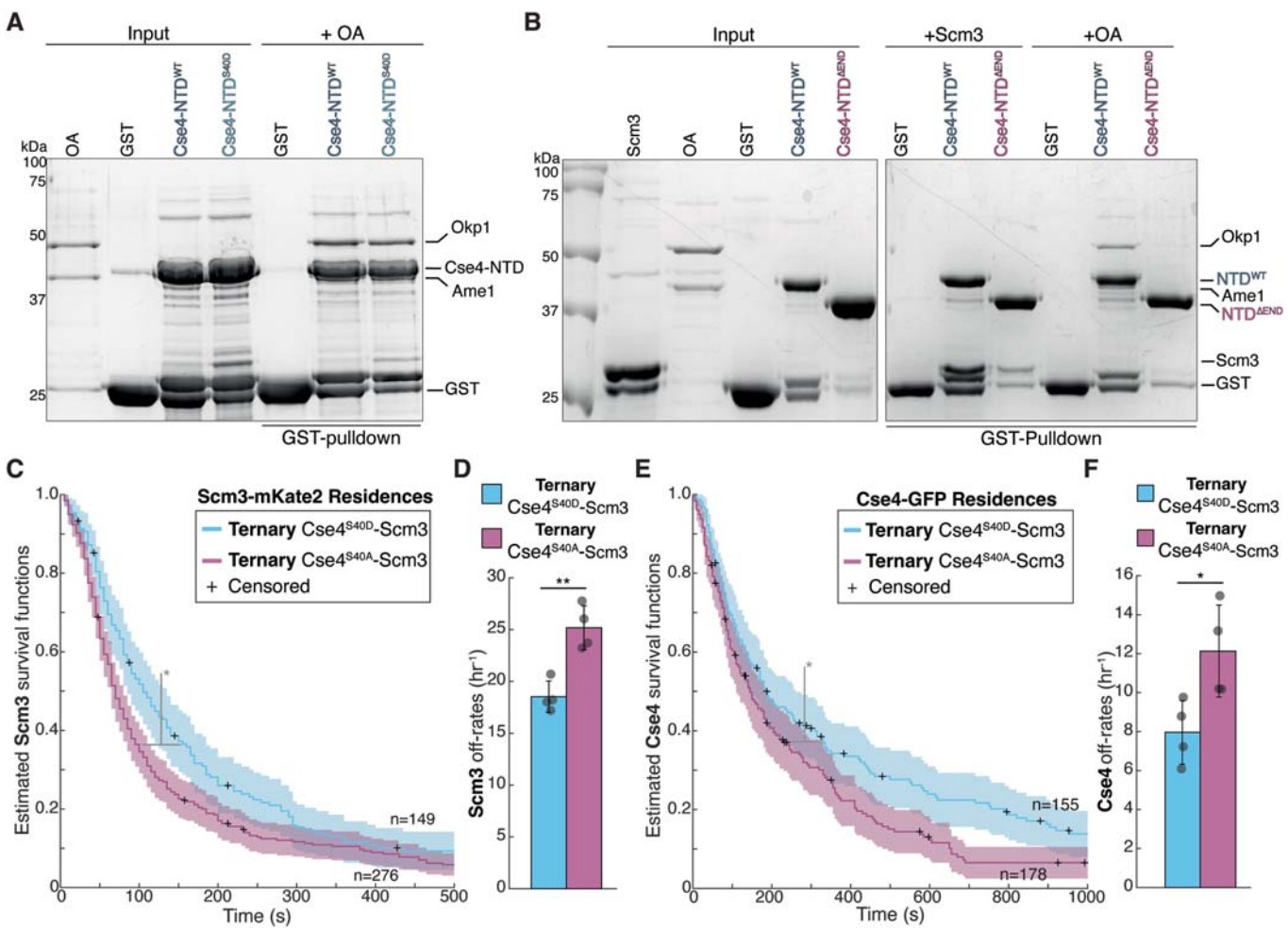

**Figure 5. Phosphorylation of Cse4 END binding domain stabilizes Cse4 on centromeric DNA.**

(A) SDS-PAGE of GST pull-down assays of immobilized GST-Cse4-NTD^WT and GST-Cse4-NTD^S40D to test binding of Okp1/Ame1. (B) SDS-PAGE of GST pull-down assays of immobilized GST-Cse4-NTD^WT and GST-Cse4-NTD^ΔEND protein fusions binding to recombinant Scm3 and Okp1/Ame1. (C) Scm3 resides on centromeres longer after colocalization with Cse4^S40D than with Cse4^S40A. Estimated survival function plots of Kaplan–Meier analysis of the lifetimes in *CSE4-S40D-GFP SCM3-mKate2* (SBY22258) extract of Ternary^S40D Scm3 residences on CEN DNA (blue—median lifetime of 104 s, *n* = 149 over four experiments of ~1000 DNA molecules using different extracts) and the lifetimes in *CSE4-S40A-GFP SCM3-mKate2* (SBY22372) Ternary^S40A Scm3 residences on CEN DNA (purple—median lifetime of 67 s, *n* = 276 over four experiments of ~1000 DNA molecules using different extracts). 95% confidence intervals indicated (shaded) right-censored lifetimes (plus icons) were included and with equivalent weighting in survival function estimates, * indicates a significant difference two-tailed *P* value of.01 as determined by log-rank test. (D) Graph indicates quantification of the estimated off-rates of Ternary^S40D and Ternary^S40A Scm3 residences on CEN DNA (18.5 h⁻¹± 2.3 h⁻¹, and 25.2 ± 2.3 h⁻¹, respectively, avg ± s.d. *n* = 395 over four experiments of ~1000 DNA molecules, ** indicates significant difference with two-tailed *P* value of.004). (E) Cse4^S40D resides on centromeres longer after colocalization with Scm3 than Cse4^S40A. Estimated survival function plots of Kaplan–Meier analysis of the lifetimes of Ternary^Scm3 Cse4^S40D residences on CEN DNA (blue—median lifetime of 184 s, *n* = 155 over four experiments of ~1000 DNA molecules using different extracts) and Ternary^Scm3 Cse4^S40A residences on CEN DNA (purple—median lifetime of 144 s, *n* = 178 over four experiments of ~1000 DNA molecules using different extracts). In all, 95% confidence intervals indicated (shaded), right-censored lifetimes (plus icons) were included with equivalent weighting in survival function estimates, * indicates significant difference two-tailed *P* value of 0.006 as determined by log-rank test. (F) Graph indicates quantification of the estimated off-rates of Ternary^Scm3 Cse4^S40D and Ternary^Scm3 Cse4^S40A residences on CEN DNA (8.0 h⁻¹± 1.6 h⁻¹, and 12.1 ± 2.4 h⁻¹, respectively, avg ± s.d. *n* = 322 over four experiments of ~1000 DNA molecules, * indicates significant difference with two-tailed *P* value of 0.03).

kinetochore assembly assays. While Cse4^S40D assembled normally (Fig. EV3A, middle), there was a reduction in Cse4^S40A centromeric association (Fig. EV3A, right). To further characterize this defect, we performed TIRFM endpoint colocalization assays. Consistent with the bulk assembly assays, Cse4^S40D-GFP localized to centromeric DNA similarly to WT (50% vs 55%, respectively), while Cse4^S40A-GFP recruitment was significantly reduced compared to WT (28% vs 55%, respectively) (Fig. EV3B). To further examine how the Cse4 END phosphorylation affects Cse4 stability on

centromeric DNA, we performed time-lapse TIRFM colocalization assays. Because the phosphorylation appeared to enhance the Scm3 interaction with Cse4, we simultaneously monitored orthogonally labeled Cse4^S40D-GFP or Cse4^S40A-GFP with Scm3-mKate2 on individual centromeric DNAs. We first asked how Scm3 stability is affected by Cse4 END domain phosphorylation within the context of the native nucleosome. Kaplan–Meier analysis of Scm3 residences on centromeric DNA that formed ternary colocalizations with either Cse4^S40D (Ternary Cse4^S40D-Scm3) or Cse4^S40A

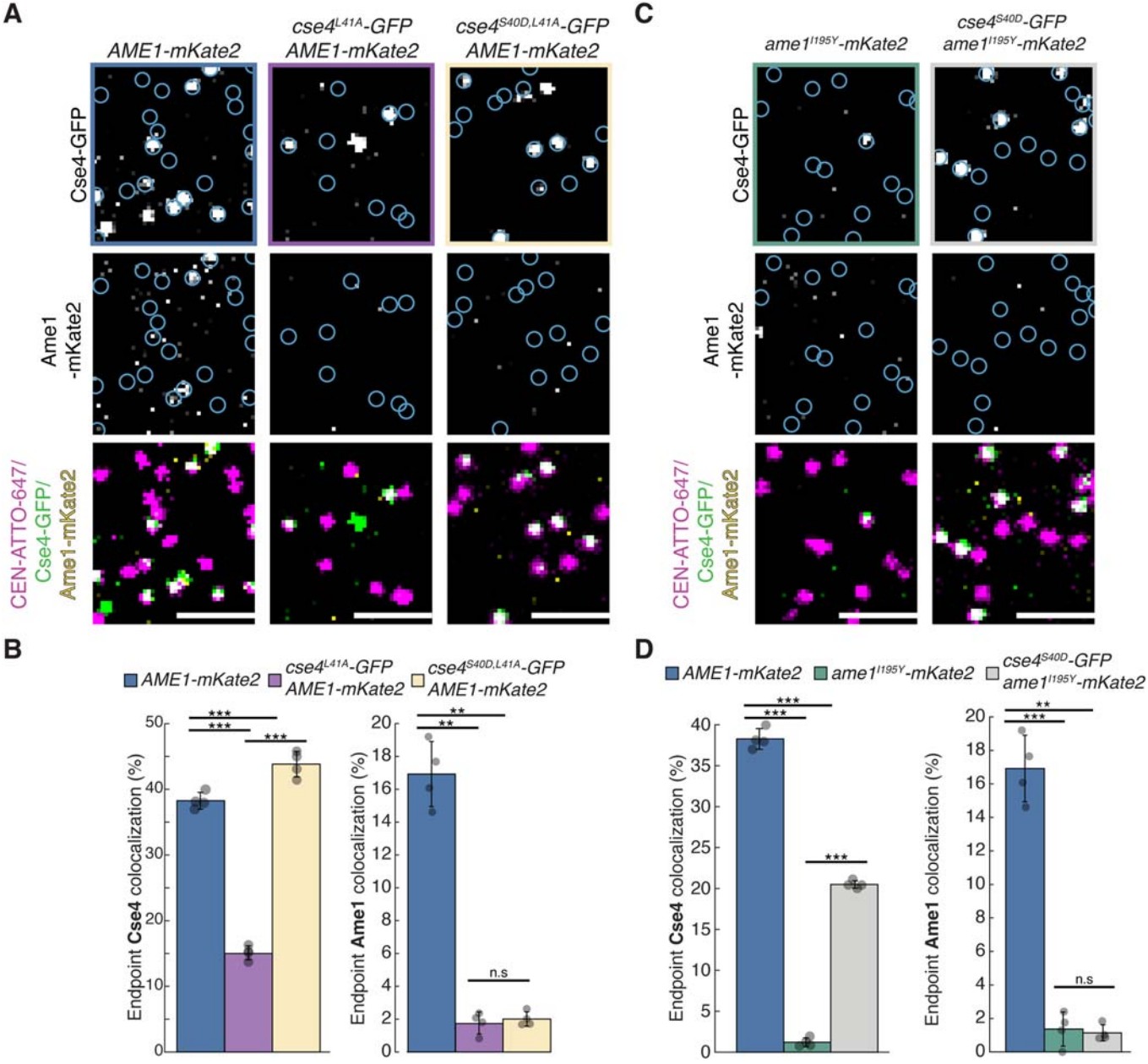

(Ternary Cse4$^{S40A}$-Scm3) revealed a significant increase in the median lifetime of Ternary Cse4$^{S40D}$ when compared to Ternary Cse4$^{S40A}$ Scm3 residences (104 s vs. 64 s, Fig. 5C). Consistent with longer Ternary Cse4$^{S40D}$-Scm3 lifetimes, off-rate analysis also revealed enhanced stability of Ternary Cse4$^{S40D}$-Scm3 when compared to Ternary Cse4$^{S40A}$-Scm3 (18.5 h$^{-1}$ vs 25.2 h$^{-1}$, Fig. 5D). We next asked whether the prolonged Scm3 binding after ternary complex formation with Cse4$^{S40D}$ also stabilized Cse4 by monitoring Cse4 residences on centromeric DNA. Kaplan–Meier analysis revealed that the median lifetimes of Ternary Cse4$^{S40D}$-Scm3 residences were significantly increased compared to Ternary Cse4$^{S40A}$-Scm3 residences on centromeric DNA (178 s vs. 144 s, Fig. 5E). This was confirmed in off-rate analysis, which showed a significantly slower average off-rate of Ternary Cse4$^{S40D}$-Scm3 versus Ternary Cse4$^{S40A}$-Scm3 residences (8.0 h$^{-1}$ ± 1.6 h$^{-1}$ and

12.1 ± 2.4 h$^{-1}$, Fig. 5F). Taken together, these analyses reveal that Cse4-S40 phosphorylation of the END region stabilizes both Scm3 and Cse4 association with centromeric DNA.

## Cse4 phosphorylation can rescue defects in Okp1/Ame1-dependent stabilization of Cse4

We next asked whether the contribution of Cse4-S40 phosphorylation to Cse4 centromere retention is independent of Okp1/Ame1. First, we tested whether the Cse4-L41 mutants that alter END association of Okp1/Ame1 affect Scm3 binding. We found that neither the Cse4-L41A or Cse4-L41D mutant affected Scm3 association to the Cse4-NTD in pull-down assays (Fig. EV4A), confirming that *cse4-L41* mutants specifically affect Okp1/Ame1 binding. We next asked whether Cse4 phosphorylation on S40

**Figure 6. Cse4 END phosphorylation rescues defects in the centromeric association of Cse4 due to disruption of Okp1/Ame1 binding to END.**

(A) Example images of TIRFM endpoint colocalization assays. Top panels show visualized Cse4-GFP on CEN DNA in *CSE4-GFP Ame1-mKate2* (SBY22244) extracts (top-left panel), *cse4$^{L41A}$-GFP AME1-mKate2* (SBY22931) extracts (top-middle panel), or *cse4$^{S40D/L41A}$-GFP AME1-mKate2* (SBY22929) extracts (top-right panel) with colocalization shown in relation to identified CEN DNA in blue circles. Middle panels show localization of Ame1-mKate in relation to identified CEN DNA in blue circles. Bottom panels show overlay of CEN DNA channel (magenta) with Cse4-GFP (green) and Ame1-mKate2 (yellow), scale bars 2 µm. (B) Graph indicates quantification of endpoint colocalization of Cse4-GFP with CEN DNA in extracts from *CSE4-GFP AME1-mKate2*, *cse4$^{L41A}$-GFP AME1-mKate2* or *cse4$^{S40D,L41A}$-GFP AME1-mKate2* genetic backgrounds (38 ± 1.3%, 15 ± 2.3%, 44 ± 2.0%, avg ± s.d. n = 4 experiments, each examining ~ 1,000 DNA molecules from different extracts, *** indicates significant difference between *CSE4-GFP AME1-mKate2* and *cse4$^{L41A}$-GFP AME1-mKate2* with two-tailed P value of 1.6E-5 and between *cse4$^{S40D,L41A}$-GFP AME1-mKate2* and *cse4$^{L41A}$-GFP AME1-mKate2* with two-tailed P value of 1.3E-6, ** indicates significant difference between *CSE4-GFP AME1-mKate2* and *cse4$^{S40DL41A}$-GFP AME1-mKate2* with two-tailed P value of 0.005.), or colocalization of Ame1-mKate2 with Cse4-GFP (17 ± 2.0%, 1.7 ± 0.4%, 2 ± 0.4%, avg ± s.d. n = 4 experiments, each examining ~1000 DNA molecules from different extracts, ** indicates significant difference between *CSE4-GFP AME1-mKate2* and *cse4$^{L41A}$-GFP AME1-mKate2* with two-tailed P value of 6.4E-4 and *CSE4-GFP AME1-mKate2* and *cse4$^{S40D,L41A}$-GFP AME1-mKate2* with two-tailed P value of 6.8E-4, n.s. indicates no significant difference between *cse4$^{S40D,L41A}$-GFP AME1-mKate2* and *cse4$^{L41A}$-GFP AME1-mKate2* with two-tailed P value of 0.3). (C) Example images of TIRFM endpoint colocalization assays. Top panels show visualized Cse4-GFP on CEN DNA in *CSE4-GFP ame1$^{I195Y}$* (SBY23105—top-left panel), or *cse4$^{S40D}$-GFP ame1$^{I195Y}$* extracts (SBY23163—top-right panel) with colocalization shown in relation to identified CEN DNA in blue circles. Middle panels show localization of Ame1-mKate in relation to identified CEN DNA in blue circles. Bottom panels show overlay of CEN DNA channel (magenta) with Cse4-GFP (green), scale bars 2 µm. (D) Graph indicates quantification of Cse4-GFP endpoint colocalization with CEN DNA in *CSE4-GFP*, *CSE4-GFP ame1$^{I195Y}$* or *cse4$^{S40D}$-GFP ame1$^{I195Y}$* extracts (38 ± 1.3%, 1 ± 0.5%, 21 ± 0.5%, avg ± s.d. n = 4 experiments, each examining ~1000 DNA molecules from different extracts, *** indicates significant difference between *CSE4-GFP AME1-mKate2* and *CSE4-GFP ame1$^{I195Y}$-mKate2* with two-tailed P value of 7.0E-7, between *CSE4-GFP AME1-mKate2* and *cse4$^{S40D}$-GFP ame1$^{I195Y}$-mKate2* with two-tailed P value of 1.2E-5 and between *cse4$^{S40D}$-GFP ame1$^{I195Y}$-mKate2* and *CSE4-GFP ame1$^{I195Y}$-mKate2* with two-tailed P value of 2.7E-9.) or colocalization of Ame1-mKate2 with Cse4-GFP (17 ± 2.0%, 1.4 ± 1.0%, 1.1 ± 0.5%, avg ± s.d. n = 4 experiments, each examining ~1000 DNA molecules from different extracts, *** indicates significant difference between *CSE4-GFP AME1-mKate2* and *CSE4-GFP ame1$^{I195Y}$-mKate2* with two-tailed P value of 1.3E-4 and *CSE4-GFP AME1-mKate2* and *cse4$^{S40D}$-GFP ame1$^{I195Y}$-mKate2* with two-tailed P value of 5.9E-4, n.s. indicates no significant difference between *cse4$^{S40D}$-GFP ame1$^{I195Y}$-mKate2* and *CSE4-GFP ame1$^{I195Y}$-mKate2* with two-tailed P value of 0.7). Note that the same WT endpoint localization data for Cse4 and Ame1 are replotted from Fig. 6B for clarity.

could rescue Okp1/Ame1 binding to the *cse4$^{L41A}$* mutant. We generated a *cse4$^{S40D,L41A}$* double mutant and performed bulk assembly assays. Cse4 levels were restored on centromeric DNA in the *cse4$^{S40D,L41A}$* mutant compared to *cse4$^{L41A}$* (Fig. EV4B). Okp1/Ame1 association was not restored, confirming that Cse4 phosphorylation on S40 does not alter the interaction with Okp1/Ame1. We quantified the rescue in TIRFM endpoint colocalization assays and found that Cse4$^{S40D,L41A}$-GFP levels were restored relative to the Cse4$^{L41A}$-GFP mutant, while the levels of Ame1-mKate2 remained disrupted in both mutants (Fig. 6A,B). In a complementary experiment, we asked whether the Cse4-S40D mutant could rescue the Cse4 centromeric localization defect in the *ame1$^{I195Y}$* mutant that disrupts Okp1/Ame1 interaction with the Cse4 END. The Cse4$^{S40D}$-GFP localization was partially rescued when compared to WT Cse4-GFP in Ame1$^{I195Y}$ mutant extracts (Fig. 6C,D).

To ensure that the rescue of Cse4 localization in the Cse4-S40D mutant did not rely on transient Okp1/Ame1 binding or other Okp1/Ame1 activity, we asked if Cse4$^{S40D}$ could still be stabilized on centromeres in the complete absence of Okp1/Ame1. We utilized a previously described auxin-inducible degron system to rapidly degrade Okp1 (*okp1-AID*) in cells containing Cse4-GFP or Cse4$^{S40D}$-GFP prior to generating extracts. We confirmed that Okp1/Ame1 depletion severely disrupted Cse4-GFP localization and it was partially rescued by the Cse4$^{S40D}$-GFP mutant in endpoint assays (Fig. EV4C). However, Cse4 levels were not rescued to the same extent as in the Cse4$^{S40D,L41A}$ mutant (Fig. 6A,B), likely because Cse4$^{L41A}$ retains residual Okp1/Ame1 binding (Fig. 3A). Taken together, our results suggest that phosphorylation of Cse4 END at S40 can promote retention of Cse4 at centromeres independently from Okp1/Ame1 binding to Cse4 END.

## Okp1/Ame1 and Cse4 phosphorylation contribute to Cse4 centromeric localization in vivo

We next analyzed the cellular consequences of the dual contributions of Cse4 END phosphorylation and Okp1/Ame1 binding to the END domain. Consistent with a previous report (Deng et al, 2023),

we found that the *cse4$^{L41A}$* mutant exhibited a significant growth defect at higher temperatures (Fig. 7A). Strikingly, the growth defect was largely rescued in the *cse4$^{S40D,L41A}$* mutant (Fig. 7A). To better understand the cause of the *cse4$^{L41A}$* mutant growth defect, we first assayed cell cycle progression at high temperature. *CSE4*, *cse4$^{L41A}$* and *cse4$^{S40D,L41A}$* mutant cells were released from a G1 arrest to 36 degrees and monitored for cell cycle progression via budding index. WT cells were large-budded 90' after release and completed cell division within 150 min (Fig. 7B). The *cse4$^{S40D,L41A}$* mutant cells progressed more slowly and became large-budded 120' after release but eventually completed cell division. In contrast, *cse4$^{L41A}$* cells accumulated as large-budded cells and never divided, suggesting activation of the spindle checkpoint (Fig. 7B). To test this, we crossed the *cse4$^{L41A}$* mutant to a *mad1Δ* checkpoint deletion mutant and found the double mutants were inviable (Fig. 7C). These data indicate that the spindle checkpoint is required for the viability of *cse4$^{L41A}$* mutant cells, strongly suggesting that there are defective kinetochore-microtubule attachments.

To further examine the potential kinetochore defects in the *cse4$^{L41A}$* cells, we imaged Cse4-GFP in the cells. In yeast, kinetochores cluster into two foci due to the pulling forces of microtubules on bioriented kinetochores (He et al, 2000; He et al, 2001; McAinsh et al, 2003; Tanaka et al, 2000). As expected, wild-type Cse4-GFP exhibited two foci in the large-budded cells at metaphase (Fig. 7D, left). In contrast, there were multiple Cse4$^{L41A}$-GFP foci, suggesting aberrant kinetochore attachments (Fig. 7D, right). Because the *cse4$^{L41A}$* mutant is synthetically lethal with a spindle checkpoint mutant, we did not assay chromosome segregation in these cells. To determine if the defective kinetochore localization was related to Cse4 stability at centromeres, we monitored Cse4 levels at centromeres relative to the fiducial centromere marker Ndc10-mCherry after shifting cells to high temperature (36 °C) under mitotic arrest for 2 h (Fig. 7E). Kinetochore clusters were identified via Ndc10 and used to quantify Cse4 centromeric intensity relative to Ndc10 on a per cluster basis and then summed per cell for comparison. Strikingly, Cse4 levels at centromeres were significantly reduced in the *cse4$^{L41A}$*

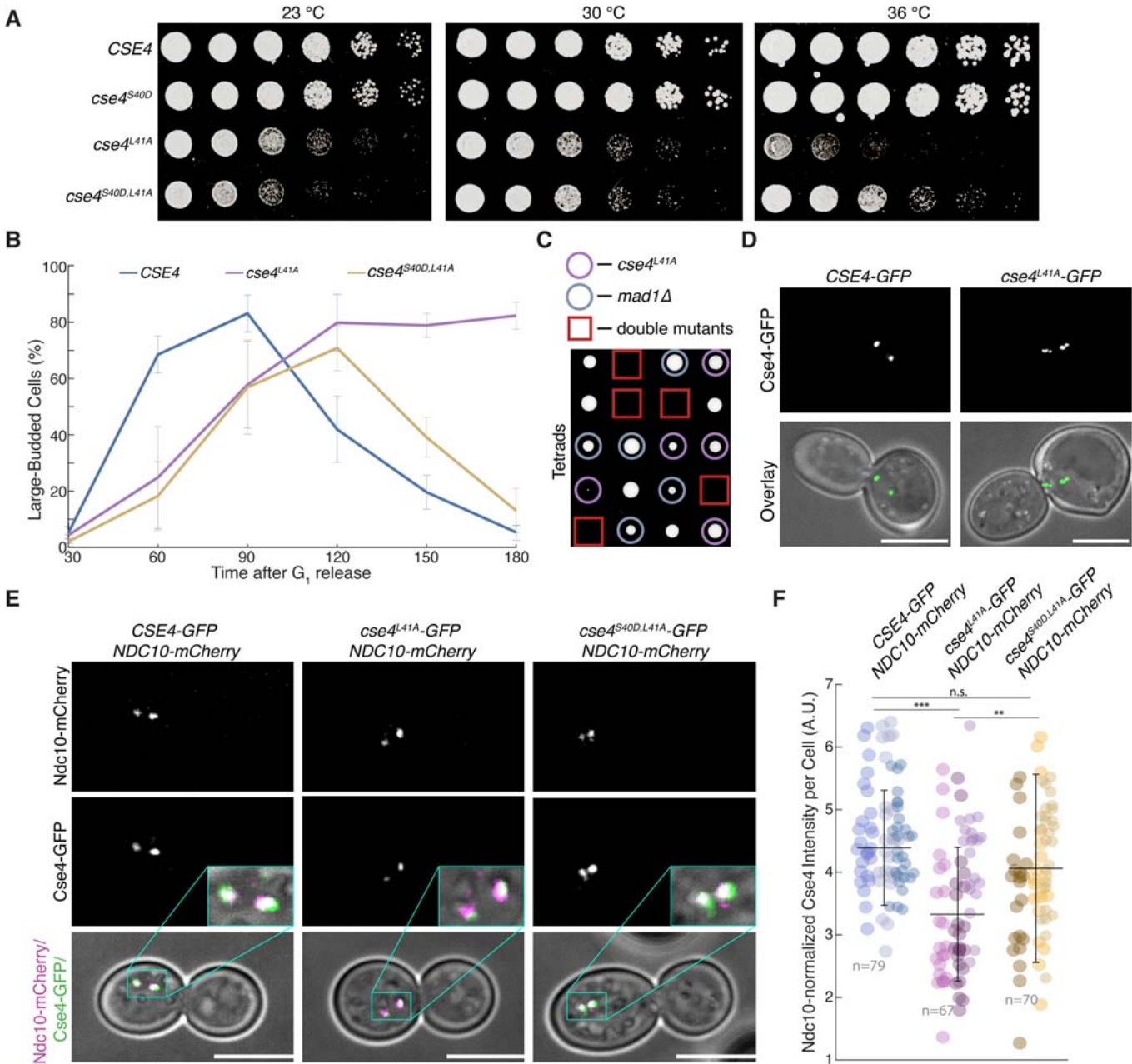

**Figure 7. Cse4-S40 phosphorylation can suppress reduced Cse4 centromeric localization caused by disruption of Okp1/Ame1 binding to END.**

(A) Fivefold serial dilutions of *CSE4-GFP* (SBY21863), *cse4^S40D^-GFP* (SBY20017), *cse4^L41A^-GFP* (SBY22811) and *cse4^S40D,L41A^-GFP* (SBY22914) strains were plated and grown on YPD for two days at 23 °C, 30 °C, and 36 °C. (B) Cse4-L41A mutant causes mitotic arrest at high temperatures. *CSE4-GFP* (blue), *cse4^L41A^-GFP* (magenta) and *cse4^S40D,L41A^-GFP* (yellow) cells were released from G₁ and fixed every 30 min, and the proportion of large-budded cells quantified. Each time point represents the percentage of budded cells for each strain (about 200 cells, avg. ± s.d., n = 3 biological replicates). (C) Haploid progeny from sporulation of *cse4^L41A^* (SBY22811)/*mad1Δ* (SBY291) heterozygous diploid, red square indicates double mutant haploid spores. Tetrads are aligned horizontally. (D) Representative images of *CSE4-GFP* (left) and *cse4^L41A^-GFP* (right) cells 90 min after G₁ release. Scale bars 5 μm. (E) Example fluorescence microscopy images of *CSE4-GFP NDC10-mCherry* (SBY21973—left), *cse4^L41A^-GFP NDC10-mCherry* (SBY23101 - middle) and *cse4^S40D,L41A^-GFP NDC10-mCherry* (SBY23474—right) cells showing visualized Ndc10-mCherry (top panels), Cse4-GFP (middle panels) and overlay of Ndc10-mCherry (magenta) and Cse4-GFP (green) on plane-polarized illumination of cell. Expanded region around kinetochores highlighted (middle panel inset). Scale bars 5 μm. (F) Graph indicates quantification of normalized centromere-associated Cse4-GFP intensity per cell of *CSE4-GFP NDC10-mCherry* (left) *cse4^L41A^-GFP NDC10-mCherry* (middle) and *cse4^S40D,L41A^-GFP NDC10-mCherry* (right) cells (4.4 ± 0.9 a.u., 3.3 ± 1.1 a.u., 4.1 ± 1.5 a.u., avg ± s.d. n = 3 experiments, each examining ~25 cells). * Indicates significant difference and n.s. indicates non-significant as determined by t test (WT:L41A P value of 1.6E-9, WT:DA P value of 0.12 and L41A:DA P value of 0.001). Each spot represents calculated intensity for one cell, different colors indicate biological replicates.

mutant and rescued to normal levels in the $cse4^{S40D,L41A}$ double mutant (Fig. 7E,F). We used the same method to analyze Cse4 levels in mitotically arrested cells depleted of Okp1/Ame1 (okp1-AID) and confirmed that Cse4 levels at centromeres were significantly reduced (Appendix Fig. S3). We also monitored Okp1/Ame1 levels using GFP-tagged Ame1 under the same conditions in $cse4^{L41A}$ and $cse4^{S40D,L41A}$ cells that were arrested in mitosis. Consistent with the in vitro assays, the centromere-associated levels of Ame1-GFP were significantly reduced in the $cse4^{L41A}$ mutant and were not rescued in $cse4^{S40D,L41A}$ mutant cells (Fig. EV5). Together, these data show that END domain contributes to Cse4 centromere localization in vivo by binding to both Okp1/Ame1 and Scm3.

# Discussion

## Stable Cse4 centromeric localization requires its END domain

While recent structural models have provided key insights into overall inner kinetochore architecture, the role of the disordered Cse4-NTD in kinetochore assembly remained unclear because most of it is not visible within the structures of the current nucleosome reconstitutions. Using single-molecule fluorescence microscopy, we discovered that the Cse4 END region is critical for Cse4 centromeric localization. To determine whether the nucleosome assembly process in conventional biochemical reconstitutions differed from de novo assembly with native components, we tested the modified centromere sequence (C0N3) used in reconstitutions in our TIRFM assay. We observed a marked reduction in Cse4 centromeric localization by the C0N3 template that replaces CDEII with a strong nucleosome positioning sequence when compared to the native centromere sequence of CEN DNA templates. These data suggest there are additional assembly pathways or contributing interactions that play a significant role in the formation of the centromeric nucleosome. These factors may be more easily distinguished in reconstitutions using cell extracts that contain these additional and potentially previously unidentified kinetochore assembly components. Our de novo kinetochore assembly assays are therefore a powerful tool to dissect pathways that are not easily detected by other assays. Using this approach, we discovered that the Cse4 END region is critical for Scm3 binding in addition to its previously established essential binding interaction with Okp1/Ame1, and that both END interactions stabilize the centromeric nucleosome.

## Okp1/Ame1 and Ipl1 phosphorylation stabilize Cse4 via the END

Cse4 END binding to Okp1/Ame1 has been well characterized and its essential function was assumed to be Okp1/Ame1 recruitment to the kinetochore. However, we found that even modest disruption of Okp1/Ame1 binding to the Cse4 END (via Cse4$^{L41A}$) significantly disrupted both Cse4 and Okp1/Ame1 centromeric localization in vitro. Consistent with this, Okp1/Ame1 also contributes to Cse4 centromeric levels in vivo. The $cse4^{L41A}$ mutant that weakens Okp1/Ame1 binding arrests in mitosis at high temperature due to spindle checkpoint activation and exhibits reduced Cse4 centromere levels. In addition, analysis of centromeric chromatin structure in vivo showed that the nucleosome structure was disrupted in the absence

of Okp1/Ame1. Together, these results suggest that Okp1/Ame1 binding to the Cse4 END domain not only recruits Okp1/Ame1 to the kinetochore, but also stabilizes the centromeric histone in the process. This END-dependent role of maintaining centromeric localization of Cse4 is consistent with the role of the N-terminal tail of CENP-A in C. elegans, where it was shown to be required for proper centromeric assembly (de Groot et al, 2021; Prosee et al, 2021). In the future, it will be important to determine whether Okp1/Ame1's interactions with additional kinetochore proteins in the CCAN further contribute to Cse4 stability at the centromere.

We also identified a role for the Scm3 chaperone in Cse4 localization via the END domain. We previously discovered that Scm3 stabilized Cse4 on centromeric DNA, and here we show it likely to be mediated at least partially through Scm3 binding to the Cse4 END, similar to the role of recruitment of KNL-2 by the N-terminal tail of CENP-A in C. elegans (de Groot et al, 2021; Prosee et al, 2021). Ipl1 phosphorylation of Cse4-S40 promotes Scm3 stability at centromeres and this END-dependent regulation is independent of Okp1/Ame1. The distinct roles of Ipl1 phosphorylation and Okp1/Ame1 in stabilizing the Cse4 nucleosome via the same END region suggest these activities might be temporally separated. Scm3 has much higher affinity for the Cse4 histone fold domain than the N-terminus (Shukla et al, 2024), so we presume Scm3 initially targets Cse4 to centromeres via the histone fold interaction. DNA wrapping would displace Scm3 from the Cse4 histone fold domain, making it accessible to bind to the N-terminus after nucleosome formation. The dual Scm3 binding sites on Cse4 may explain its exchange at kinetochores throughout the cell cycle (Wisniewski et al, 2014). Although Okp1/Ame1 binds to the Cse4 END with higher affinity than Scm3, it is unclear whether they simultaneously bind. In addition, the interaction between Scm3 and the Cse4 END is much different than the END domain association with Okp1/Ame1. The END domain is intrinsically disordered but becomes helical when bound to Okp1/Ame1 (Deng et al, 2023). In contrast, the END interaction with Scm3 does not impart order (Shukla et al, 2024), suggesting that there may be additional stabilizing interactions in vivo. In the future, it will be important to further understand the structural nature of the Scm3 interaction with the END and whether there is a temporal order to Scm3 and Okp1/Ame1 stabilization of Cse4 during kinetochore assembly.

## Ipl1 phosphorylation of the Cse4 END can compensate for weakened Okp1/Ame1 recruitment

Previous work suggested that Cse4 overexpression can bypass the essential requirement for the N-terminus of the histone variant or the Scm3 chaperone (Camahort et al, 2007; Morey et al, 2004). A possible explanation for these findings is that Cse4 overexpression drives its stability at the centromere and therefore allows cells to live without the additional stabilizing mechanisms we have identified. Consistent with this, Ipl1 phosphorylation of the Cse4 END partially rescued defects in Cse4 centromeric localization caused by the Cse4$^{L41A}$ Okp1/Ame1 binding mutant both in vitro and in vitro. We propose that this rescue was driven by Scm3 binding as Okp1/Ame1 levels were not rescued in the Cse4$^{S40D,L41A}$ double mutant. Moreover, the Cse4$^{S40D}$-dependent rescue in Cse4 localization could be maintained even in the most extreme case, where Okp1/Ame1 was absent during kinetochore assembly

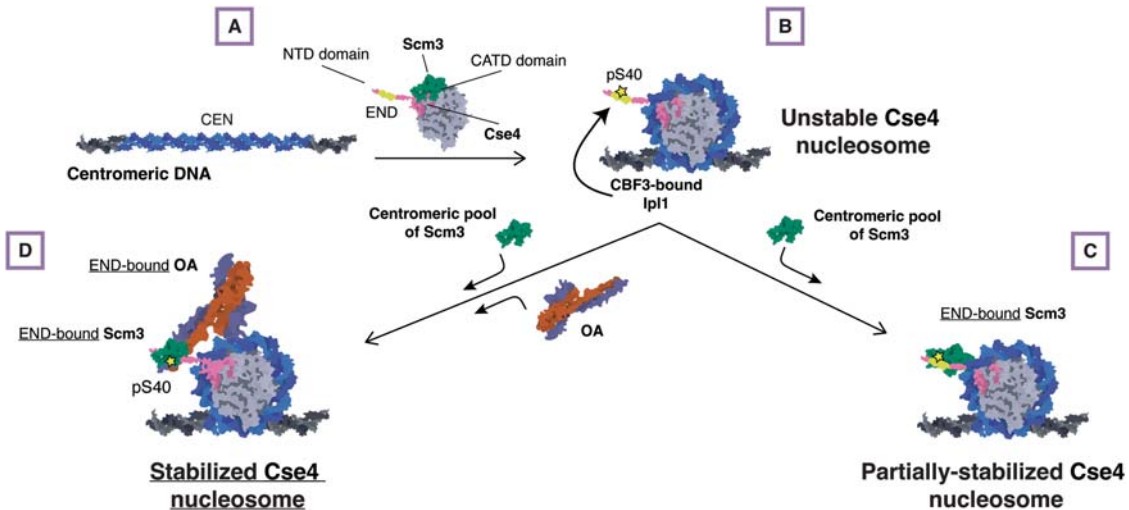

**Figure 8.  Model for stabilization of Cse4 at centromeres.**

Schematic diagram of the proposed model of Cse4 nucleosome formation that relies on END recruitment of the essential chaperone Scm3 and the essential CCAN component Okp1/Ame1 for stability prior to kinetochore assembly. **(A)** CATD-bound Scm3 targets Cse4 to the centromere. **(B)** CATD-bound Scm3 is released when DNA wraps to form nucleosome and centromere-associated (CBF3) Ipl1 phosphorylates Cse4 at S40. **(C)** Scm3 alone can bind to the Cse4-NTD, promoted by END phosphorylation at S40, to partially stabilize Cse4 nucleosome. **(D)** Recruitment of both Okp1/Ame1 and Scm3 to Cse4 END are required to stabilize Cse4 on centromeric DNA to enable kinetochore assembly.

(Fig. EV4B). Together, these results highlight previously unknown functions for Scm3 in promoting Cse4 stability on centromeric DNA, furthering our understanding of the unique Scm3 chaperone protein which is part of a larger family of histone chaperones whose set of cellular functions continue to expand (Hammond et al, 2017). Moreover, these additional Scm3 functions depend on interactions at the kinetochore that are not via the histone fold domain, suggesting additional chaperone functions important for kinetochore assembly.

## Regulation of Cse4 localization

Our finding that phosphorylation of Cse4-S40 promotes Cse4 localization highlights a previously unknown regulatory point of kinetochore assembly. While it had been previously shown that Ipl1 phosphorylation of Cse4 played a role in chromosome segregation (Boeckmann et al, 2013), the underlying mechanisms were not known. Here, we identified the function of one (S40) of the four previously characterized sites (S22, S33, S40, and S105). We note that this phosphorylation event is not essential, likely due to the multiple mechanisms we've identified that ensure Cse4 localization and stability in vivo. In addition, given the number of kinetochore substrates that Ipl1 phosphorylates, it is likely that Ipl1 has additional key substrates that regulate Cse4 stability and kinetochore assembly. Consistent with this, the Cse4-S40 phosphorylation site does not fit the canonical Ipl1 consensus motif, suggesting there may be many more Ipl1 phosphorylation events at the kinetochore than currently known. The Mif2 and Mad3 proteins have Ipl1 sites that also do not match the consensus so it will be important to identify the full suite of kinetochore sites phosphorylated by Ipl1 (Hinshaw et al, 2023; King et al, 2007). We propose that Cse4-S40 phosphorylation occurs directly at centromeres because Ipl1 associates with the Ndc10 protein that binds to the centromere.

Although Ipl1 is most well-known for its role in error correction, our work has identified another function for Ipl1 in kinetochore assembly. It was previously established that Ipl1 facilitates kinetochore assembly by relieving autoinhibition of the interaction between Mif2 and the MIND complex by phosphorylation of the Dsn1 protein (Akiyoshi et al, 2013; Dimitrova et al, 2016). The Ipl1 regulation of Dsn1 is a conserved kinetochore assembly mechanism (Bonner et al, 2019; Kim and Yu, 2015; Petrovic et al, 2016), and it will be interesting to determine whether the regulation of the centromeric histone variant is conserved. CENP-A is known to be phosphorylated by Aurora B in other organisms (Zeitlin et al, 2001), but it isn't yet clear whether the chaperone HJURP binds to a second region in CENP-A.

It is perplexing that the centromere is a poor nucleosome assembly template in conventional biochemical reconstitutions. The inherent instability of the yeast centromeric nucleosome has precluded kinetochore reconstitutions using complete native centromere sequences. Here, we have identified multiple previously unknown mechanisms that ensure the stabilization of the yeast centromeric nucleosome. We propose that yeast use a multistep "licensing" assembly mechanism to ensure that chromosomes assemble a single centromeric nucleosome (Fig. 8). We presume that the initial Cse4 association with centromeres is driven by Scm3 binding to the CATD within the histone fold domain to target Cse4 to the centromere. Wrapping of the centromeric DNA around the histones would displace Scm3. Once Cse4 is at the centromere, Ipl1 would be in the proximity of the Cse4 END and phosphorylate it to promote Scm3 binding to the N-terminus to promote partial stabilization of the nucleosome. The END domain would also bind to Okp1/Ame1, which is needed to fully stabilize the centromeric nucleosome. Because Ipl1 is associated with the CBF3 complex that exclusively binds to the CDEIII centromere element, this event could not occur in euchromatin and would therefore help prevent

stable ectopic Cse4 deposition to maintain chromosomal integrity during mitosis. This complex regulatory schema relies on the unique Cse4 END region, which may be a consequence of the specific point centromere architecture present in budding yeast where a single centromeric nucleosome is sufficient to assemble the kinetochore. The stringent centromere sequence composition requires the coordination of several kinetochore proteins to form centromeric nucleosomes, ensuring the specificity of centromeric histone deposition which is crucial to cell division and organismal survival. Further study of kinetochore assembly pathways under differing cellular contexts is critical for understanding how kinetochores form and function in cells.

# Methods

### Reagents and tools table

| Reagent/resource | Reference or source | Identifier or catalog number |
|---|---|---|
| **Experimental models** | | |
| *S. cerevisiae cell lines* | This study | Appendix Table S1 |
| *Rosetta2(DE3) E. coli* | Sigma Aldrich | 71397-3 |
| **Recombinant DNA** | | |
| Plasmids for yeast strain generation and recombinant protein expression. | This study | Appendix Table S2 |
| **Antibodies** | | |
| α-Cse4 | Pinsky et al, 2003 | 9536 |
| α-Ndc10 | Lang et al, 2018 | OD1 |
| α-Ame1 | This study, Genscript | 2181 |
| α-V5 | Invitrogen | R96025 |
| α-Rabbit IgG peroxidase-linked species-specific whole antibody (from donkey) Secondary Antibody | Cytiva | NA9341ML |
| **Oligonucleotides and other sequence-based reagents** | | |
| Oligonucleotides | This study | Appendix Table S3 |
| **Chemicals, enzymes, and other reagents** | | |
| Protocatechuic acid | Sigma Aldrich | PHL89766 |
| Protocatechuate 3,4-Dioxygenase from Pseudomonas sp. | Sigma Aldrich | P8279 |
| 6-hydroxy-2,5,7,8-tetramethylchroman-2-carboxylic acid | Sigma Aldrich | 238813 |
| BIO-PEG-SVA, MW 5,000 and MPEG-SVA, MW 5,000 | Laysan Bio | BIO-PEG-SVA-5K-100MG & MPEG-SVA-5K-1g |
| Avidin DN | Vector Laboratories | A-3100-1 |
| Vectabond Tissue Section Adhesive | Vector Laboratories | SP-1800 |

| Reagent/resource | Reference or source | Identifier or catalog number |
|---|---|---|
| Phusion High-Fidelity DNA Polymerase | New England Biolabs | M0530S |
| SuperSignal™ West Dura Extended Duration Substrate | Thermo Fisher | 34075 |
| **Software** | | |
| Fiji | https://fiji.sc | |
| MATLAB | https://matlab.mathworks.com | |
| ASMCA | Popchock et al, 2023 https://github.com/FredHutch/Automated-Single-Molecule-Colocalization-Analysis | |
| **Other** | | |
| Nikon TE-2000 inverted RING-TIRF microscope | Nikon | |
| DeltaVision Ultra | GE Healthcare | |
| ChemiDoc Imager | Bio-Rad | |

## Yeast methods

The *S. cerevisiae* strains used in this study are listed in Appendix Table S1 and are derived from SBY3 (W303). Strains were generated using standard genetic crosses, media and PCR-based tagging techniques (Longtine et al, 1998; Schneider et al, 1995; Sherman, 2002). The plasmids and primers used to generate strains are listed in Appendix Table S2 and Appendix Table S3, respectively. Single point mutants (*cse4-S40D*, *cse4-S40A*, and *cse4-L41A*) were introduced into endogenous alleles using standard yeast CRISPR techniques (DiCarlo et al, 2013). *CSE4-GFP* and all mutant derivatives were internally tagged at residue 80 with eGFP including linkers on either side (pSB1617) and then expressed from its native promoter at the endogenous locus. *CSE4-mCherry* is equivalently constructed with mCherry in place of eGFP (pSB3271) and expressed similarly (SBY20858). Endogenous genes that were altered to include fluorescent protein alleles (eGFP, mCherry or mKate2) tags or auxin-inducible degrons (-IAA7) were constructed by standard PCR-based integration techniques (Longtine et al, 1998) and confirmed by PCR. All liquid cultures were grown in yeast peptone dextrose-rich (YPD) media. For kinetochore assembly assays, cells were arrested in mitosis by diluting log phase cultures into benomyl-containing liquid media to a final concentration of 30 μg/mL and grown for another three hours until at least 90% of cells were large-budded. For strains with auxin-inducible degron (AID) alleles (*ipl1-AID*, *okp1-AID*), cultures were treated with 500 μM indole-3-acetic acid (IAA, dissolved in DMSO) for the final 60 min of mitotic arrest as described previously (Lang et al, 2018; Miller et al, 2016; Nishimura et al, 2009). For strains with galactose inducible alleles, cultures were grown in raffinose and mitotically arrested in raffinose and benomyl-containing liquid media and then treated with 4% galactose for the final 60 min of mitotic arrest. Growth assays were performed by diluting log phase

cultures to $OD_{600 \sim 1.0}$ and plating a 1:5 serial dilution series on YPD and growing the spotted cells at 23 °C, 30 °C, and 36 °C. For budding index assays, log phase liquid cultures were arrested with alpha-factor for three hours at room temperature and then washed and released into the cell cycle at 36 °C. Cells were collected and fixed every 20 min and alpha-factor was added once ~50% cells appeared large-budded to prevent progression into the following cell cycle. Cells were visualized on concanavalin A-functionalized coverslips using plane-polarized light with a 60X objective on an inverted Nikon light microscope. At each respective time point and for each respective strain, ~200 cells were counted for large-budded appearance.

## Whole-cell extract preparation for kinetochore assembly assays

Kinetochore assembly assays were performed as described in (Lang et al, 2018; Popchock et al, 2023). Briefly, cells were grown in liquid YPD media to log phase and arrested in mitosis in 500 mL total volume and then harvested by centrifugation. All subsequent steps were performed on ice with 4 °C buffers. Cells were washed once with dH₂O with 0.2 mM PMSF, then once with Buffer L (25 mM HEPES pH 7.6, 2 mM MgCl₂, 0.1 mM EDTA pH 7.6, 0.5 mM EGTA pH 7.6, 0.1% NP40, 175 mM K-Glutamate, and 15% Glycerol) supplemented with protease inhibitors (10 mg/ml leupeptin, 10 mg/ml pepstatin, 10 mg/ml chymostatin, 0.2 mM PMSF) and 2 mM DTT. Cell pellets were then snap-frozen in liquid nitrogen and lysed using a Freezer/Mill (SPEX SamplePrep), using ten rounds that consisted of 2 min of bombarding the pellet at 10 cycles per second, then cooling for 2 min. The subsequent powder was weighed and then resuspended in Buffer L according to the following calculation: weight of pellet (g) × 1.5 = number of mL of Buffer L. Resuspended cell lysate was thawed on ice and clarified by centrifugation at 16,100 × g for 30 min at 4 °C, the protein-containing layer was extracted with a syringe and then aliquoted and snap-frozen in liquid nitrogen. The resulting soluble whole-cell extracts (WCE) generally had a concentration of 60–80 mg/mL. The pellets, powder, and WCE were stored at −80 °C.

## Preparation of DNA templates and dynabeads for assembly assays

As described previously (Popchock et al, 2023), plasmid pSB963 was used to generate the WT CEN DNA templates, and pSB972 was used to generate the CEN^mut template used in this study. PCR products were purified using the Qiagen PCR Purification Kit. In the case of bulk assembly assays, purified CEN DNA was conjugated to Streptadivin-coated Dynabeads (M-280 Streptavidin, Invitrogen) for 2.5 hr at room temperature, using 1 M NaCl, 5 mM Tris-HCl (pH 7.5), and 0.5 mM EDTA as the binding and washing buffer. For single-molecule TIRFM assays, purified CEN DNA was diluted in TE to a final concentration ~100 pM.

## Bulk assembly assays and immunoblotting

Bulk kinetochore assembly was performed with whole-cell extract and WT CEN or CEN^mut DNA conjugated to Streptadivin-coated Dynabeads as previously described (Lang et al, 2018). Briefly, 0.5 mL of whole-cell extract and 25 μl of DNA-coated M-280 Dynabeads (250 bp WT CEN)

were incubated at room temperature for either 90 min to allow complete kinetochore assembly or incubated at room temperature and stopped at 20 min, 40 min, and 80 min by washing 3× and then resuspending in 1× Laemmli sample buffer (Bio-Rad). Samples were then resolved via 10% SDS-PAGE. Proteins were then transferred from SDS-PAGE gels onto 0.22 μM cellulose paper, blocked at room temperature with 4% milk in PBST, and incubated overnight at 4 °C in primary antibody. Antibody origins and dilutions in PBST were as follows: α-Cse4 (9536 (Pinsky et al, 2003); 1:500), α-Ndc10 (OD1 (Lang et al, 2018); 1:15,000). The anti-Ame1 antibody was generated in rabbits against recombinant protein by Genscript. The company provided affinity-purified antibodies for each protein that we validated by immunoprecipitation of the respective target protein from yeast strains with an endogenously V5-tagged Ame1 allele of the target protein, followed by confirmation of antibody recognition a protein of the correct molecular weight that was also recognized by α-V5 antibody (Invitrogen; R96025; 1:5000). We subsequently used the Ame1 antibody at the following dilution (2181; 1:5000). Secondary antibodies were validated by the same methods as the primary antibodies as well as with negative controls lacking primary antibodies to confirm specificity. Blots were then washed again with PBST and incubated with secondary antibody at room temperature. Secondary antibody was α-rabbit (NA934), horseradish peroxidase-conjugated purchased from GE Healthcare and used at 1:1000 dilution in 4% milk in PBST. Blots were then washed again with PBST and ECL substrate (Thermo Scientific) was used to visualize the proteins on a ChemiDoc Imager (Bio-Rad). Uncropped and unprocessed scans of blots are provided in the Source Data file.

## Recombinant protein expression and purification

All constructs were expressed in BL21 Rosetta2 cells (EMD Millipore) as follows: cells were grown in TB media to an OD of ~0.6 and then induced with 0.1 M IPTG at 18 °C for 16 h. After induction, cells were harvested and then resuspended in binding buffer (25 mM HEPES, 175 mM KGlut, 6 mM MgCl₂, 15% glycerol, and 0.1% NP40, pH 7.6) supplemented with protease inhibitors and then lysed via sonication. The extract was spun at 14k RCF for 45 min to pellet debris. For HIS-tagged constructs (Scm3, Okp1/Ame1), clarified cell supernatant was supplemented with 10 mM immidazole and then incubated with HisPur cobalt resin (Thermo Fisher Scientific) for 40 min at 4 °C and then washed with 20× resin bed volume of binding buffer supplemented with 20 mM immidazole. Proteins were then eluted with binding buffer supplemented with 150 mM imidazole, and fractions were analyzed via SDS-PAGE. The purest elution fractions were pooled, and buffer-exchanged to remove immidazole. Protein aliquots were snap-frozen in liquid nitrogen. For GST-tagged proteins (Cse4-NTD constructs), clarified cell supernatant was incubated with Glutathione MagBeads (GenScript) for 40 min at 4 °C and then washed with 20× bead volume of binding buffer. Washed beads were then analyzed via SDS-PAGE and diluted in binding buffer to a final concentration of ~150 μM. Protein aliquots were snap-frozen in liquid nitrogen.

## In vitro binding assays

GST-Cse4-NTD constructs (WT, S40D, L41A, L41D, ΔEND) were diluted to ~15 μM in binding buffer (25 mM HEPES, 175 mM KGlut, 6 mM MgCl₂, 15% glycerol, and 0.1% NP40, pH 7.6) with

~15 µM of Ame1-6xHIS/Okp1 or Scm3-6xHIS and then mixed via rotation at 25 °C for 30 min. Resin was then washed at least three times with binding buffer to remove unbound protein, and bound protein was then eluted with Laemmli Sample Buffer and boiling. Input and bead-bound proteins were analyzed by SDS-PAGE visualized by Coomassie brilliant blue staining. For Scm3 binding experiments, titrations of GST-Cse4-NTD (WT and S40D) and GST constructs were diluted to ~15 µM in binding buffer containing 30 µM, 15 µM or 7.5 µM Scm3-6xHIS (2:1, 1:1, 1:2 Cse4-NTD:Scm3 respectively) and then mixed via rotation at 25 °C for 30 min. Resin was then washed and analyzed by SDS-PAGE as before. To determine relative binding of Scm3 to Cse4-NTD constructs, SDS-PAGE Scm3-6xHIS protein band total intensity was determined using FIJI (Schindelin et al, 2012) and normalized to total intensity of Cse4-NTD protein band in the corresponding lane. This experiment was repeated three times and binding ratios were averaged between repeats of input concentrations.

## Single-molecule TIRFM slide preparation

Coverslips and microscope slides were ultrasonically cleaned and passivated with PEG as described previously (Larson and Hoskins, 2017; Larson et al, 2013). Briefly, ultrasonically cleaned slides were treated with vectabond (Vector Laboratories) prior to incubation with 1% (w/v%) biotinylated mPEG-SVA MW-5000K/mPEG-SVA MW-5000K (Lysan Bio) in flow chambers made with double-sided tape. Passivation was carried out overnight at 4 °C. After passivation, flow chambers were washed with Buffer L and then incubated with 0.3 M BSA/0.3 M Kappa Casein in Buffer L for 5 min. Flow chambers were washed with Buffer L and then incubated with 0.3 M Avidin DN (Vector Laboratories) for 5 min. Flow chambers were then washed with Buffer L and incubated with ~100 pM CEN DNA template for 5 min and washed with Buffer L. For endpoint colocalization assays, slides were prepared as follows: Flow chambers were filled with 100 µL of WCE containing protein(s) of interest via pipetting and wicking with filter paper. After addition of WCE, slides were incubated for 90 min at 25 °C and then WCE was washed away with Buffer L. Flow chambers were then filled with Buffer L with oxygen scavenger system (Aitken et al, 2008) (10 nM PCD/2.5 mM PCA/1 mM Trolox) for imaging. For time-lapse colocalization assays, slides were prepared as follows: On the microscope, 100 µL WCE spiked with oxygen scavenger system was added to flow chamber via pipetting followed by immediate image acquisition.

## Single-molecule TIRFM colocalization assays image collection and analysis

The TIRFM colocalization assays were performed as previously described (Popchock et al, 2023). Briefly, images were collected on a Nikon TE-2000 inverted RING-TIRF microscope with a 100× oil immersion objective (Nikon Instruments) with an Andor iXon X3 DU-897 EMCCD camera. Images were acquired at 512 px × 512 px with a pixel size of 0.11 µm/px at 10 MHz. Atto-647 labeled CEN DNAs were excited at 640 nm for 300 ms, GFP-tagged proteins were excited at 488 nm for 200 ms, and mCherry/mKate2-tagged proteins were excited at 561 nm for 200 ms. For endpoint colocalization assays, single snapshots of all channels were acquired. For real-time colocalization assays images in 561 nm

channel and 488 nm channel were acquired every 5 s with acquisition of the DNA channel (647 nm) every 1 min for 45 min total (541 frames) using Nikon Elements acquisition software. Single endpoint images were analyzed using ComDet v.0.5.5 plugin for ImageJ (https://github.com/UU-cellbiology/ComDet) to determine colocalization and quantification between DNA channel (647 nm) and GFP (488 nm) and mCherry (561 nm) channels. Results were quantified and plotted using MATLAB (The Mathworks, Natick, MA). Adjustments to example images (contrast, false color, etc.) were made using FIJI (Schindelin et al, 2012) and applied equally across entire field of view of each image.

Real-time colocalization assays were performed as described (Popchock et al, 2023). Briefly, a custom-built image analysis pipeline was built in MATLAB (R2019b) to extract DNA-bound intensity traces for the different fluorescent species, to convert them into ON/OFF pulses and to generate the empirical survivor function data (Popchock et al, 2023). The image dataset was drift-corrected using either fast Fourier Transform cross-correlations or translation affine transformation depending on the severity of the drift. DNA spots were identified after binarizing the DNA signal using global background value as threshold, as well as size and time-persistency filtering. Mean values of z-normalized fluorescent markers intensities were measured at each DNA spot at each time frame, and local background was subtracted. Z-normalized traces were then binarized to ON/OFF pulses by applying a channel-specific, manually adjusted threshold value unique to all traces in a given image set. Pulse onsets, durations and overlaps between channels were then derived. For plotting clarity, z-normalized traces shown in the figures were zero-adjusted so that the baseline signal lies around zero. Pulses in ON state at the end of the image acquisition (right-censored) were annotated and included as unweighted in lifetime estimates. Kaplan–Meier analysis and log-rank tests were performed in MATLAB (R2023a). Adjustments to example plot images (contrast) as well as generation of example plot source movies were made using FIJI (Schindelin et al, 2012).

## Live-cell microscopy and image analysis

Live-cell microscopy was performed as described previously (Herman et al, 2020) with slight modification. Briefly, fixed cell images were acquired on a DeltaVision Ultra deconvolution high-resolution microscope (GE Healthcare) equipped with a 100×/1.42 PlanApo N oil immersion objective (Olympus) and a 16-bit sCMOS detector. Cells were imaged in Z-stacks through the entire cell using 0.2-µm steps using plane-polarized light, 488 nm, and 568 nm illumination. All images were deconvolved using standard settings. Z projections of the maximum signal in all channels were exported as TIFFs for analysis in FIJI (Schindelin et al, 2012). 568 nm images of Ndc10-mCherry were used to identify centromeric region puncta in cells. The signal intensity within these regions was quantified for the 568 nm channel and then for the corresponding 488 nm channel images on a per cell basis. Total intensity in the 488 nm channels was normalized to total 568 nm intensity on a per cell basis and averaged between ~20 cells per biological replicate (total of three biological replicates per strain imaged). Representative images displayed from these experiments are projections of the maximum pixel intensity across all Z images. Adjustments to example cell images (contrast) were made using FIJI (Schindelin et al, 2012).

## Fiber-Seq analysis of *Saccharomyces cerevisiae* genomic DNA

In-house Hia5 preparation and Fiber-seq were performed as described (Stergachis et al, 2020), with some adaptation for yeast cultures. Briefly, 10 mL of yeast cells (WT, *okp1-AID*, and *cse4-AID*) were grown in YPD media to mid-log phase and were treated with 500 µM indole-3-acetic acid (IAA, dissolved in DMSO) for two hours before being harvested by centrifugation. Cells were washed once with cold dH$_2$O and resuspended in cold KPO4/Sorbitol buffer (1 M Sorbitol, 50 mM Potassium phosphate pH 7.5, 5 mM EDTA pH 8.0) supplemented with 0.167% β-Mercaptoethanol. Cells were spheroplasted by addition of 0.15 ug/mL final concentration of Zymolyase T100 (Amsbio) and incubated at 23 °C for 15 min on a roller drum. Spheroplasts were pelleted at 1200 rpm for 8 min at 4 °C, washed twice with cold 1 M Sorbitol, and resuspended in 58 uL of Buffer A (1 M Sorbitol, 15 mM Tris-HCl pH 8.0, 15 mM NaCl, 60 mM KCl, 1 mM EDTA pH 8.0, 0.5 mM EGTA pH 8.0, 0.5 mM Spermidine, 0.075% IGEPAL CA-630). Spheroplasts were treated with 1 µL of Hia5 MTase (200U) and 1.5 µL of 32 mM S-adenosylmethionine (NEB) for 10 min at 25 °C. Reaction was stopped by addition of 3 µL of 20% SDS (1% final concentration) and high molecular weight DNA was purified using the Promega Wizard® HMW DNA extraction kit (A2920). HMW Fiber-seq modified gDNA was sheared using a g-TUBE (520079) and PacBio SMRTbell libraries were then constructed and sequenced as described (preprint: Bohaczuk et al, 2024). Circular consensus sequence reads were generated from raw PacBio subread files and processed as described (preprint: Bohaczuk et al, 2024). Reads were mapped to the April 2011 sacCer3 yeast reference genome. Nucleosomes were then defined using the default parameters of fibertools (Jha et al, 2024). To control for minor technical differences in the overall m6A methylation rate between each sample, reads from each sample were subsetted such that the final distribution of per-read methylation rate was identical between each sample (https://github.com/StergachisLab/match-distribution). Fibers overlapping with the center of each sixteen centromeres +/− 500 bp were extracted. The nucleosome density was calculated by counting the number of nucleosomes that overlap with each base pair of the region of interest divided by the number of fibers overlapping with that position. The nucleosome density of all sixteen centromeric regions are averaged and plotted in Fig. 1B.

### Statistical tests

The significance between colocalization percentages in endpoint assays and in-cell normalized fluorescence intensity was determined by two-tailed unpaired *t* tests. As described previously (Popchock et al, 2023), right-censored lifetimes were included in an unweighted Kaplan–Meier analysis to estimate survival functions, censored events typically comprised <5% of observed lifetimes. Survival functions including 95% confidence intervals were plotted using KMplot (Cardillo G. 2008) and *P* values for comparing the Kaplan–Meier survival plots are provided in the figure legends and were computed using the log-rank test within Logrank (Cardillo G. 2008) as performed in (Popchock et al, 2023). Different lifetime plots were considered significantly different with a *P*

value less than 0.05. Significance between off-rates as well as between cellular intensity values were determined by two-tailed unpaired *t* tests.

## Data availability

Custom software written in MATLAB (R2021a) was used for TIRF colocalization residence lifetime analysis and plot generation. The source code is publicly available at https://github.com/FredHutch/Automated-Single-Molecule-Colocalization-Analysis. The sequencing data generated in this study have been deposited in the NCBI Sequence Read Archive (SRA) database under the accession number PRJNA1189155. All primary datasets and associated data used for analysis in all figures are available at Zenodo, https://zenodo.org/ and assigned the identifier (https://doi.org/10.5281/zenodo.14053307). All primary data reported in this paper will be shared by the lead contact upon request.

The source data of this paper are collected in the following database record: biostudies:S-SCDT-10_1038-S44318-024-00345-5.

## Peer review information

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

## Acknowledgements

The authors thank Shane Neph for his help with the Fiber-seq bioinformatics analysis and Arshad Desai for antibodies. The authors also thank members of the SB and CLA labs for critical reading of the manuscript. All imaging was performed at the Fred Hutchinson Cancer Center Cellular Imaging Core (supported by the NIH P30 CA015704 grant to the Fred Hutch/University of Washington Cancer Consortium), and we thank Hoku West-Foyle and Lena Schroeder for their experimental help. This work was supported by postdoctoral fellowship NIH F32GM136010 to ARP, by NIH R35GM134842 to CLA, by NIH 1DP5OD029630 to ABS, and by NIH R35 GM149357 to SB. ABS holds a Career Award for Medical Scientists from the Burroughs Wellcome Fund and is a Pew Biomedical Scholar. SB is an investigator of the Howard Hughes Medical Institute.

## Author contributions

**Andrew R Popchock**: Conceptualization; Resources; Data curation; Formal analysis; Funding acquisition; Investigation; Methodology; Writing—original draft; Project administration; Writing—review and editing. **Sabrine Hedouin**: Conceptualization; Data curation; Formal analysis; Investigation; Methodology; Writing—review and editing. **Yizi Mao**: Conceptualization; Resources; Data curation; Formal analysis; Investigation; Methodology. **Charles L Asbury**: Conceptualization; Resources; Supervision; Funding acquisition; Methodology; Writing—review and editing. **Andrew B Stergachis**: Conceptualization; Resources; Supervision; Funding acquisition; Methodology; Project administration; Writing—review and editing. **Sue Biggins**: Conceptualization; Resources; Formal analysis; Supervision; Funding acquisition; Investigation; Visualization; Methodology; Writing—original draft; Project administration; Writing—review and editing.

Source data underlying figure panels in this paper may have individual authorship assigned. Where available, figure panel/source data authorship is listed in the following database record: biostudies:S-SCDT-10_1038-S44318-024-00345-5.

## Disclosure and competing interests statement

The authors declare no competing interests.

# Expanded View Figures

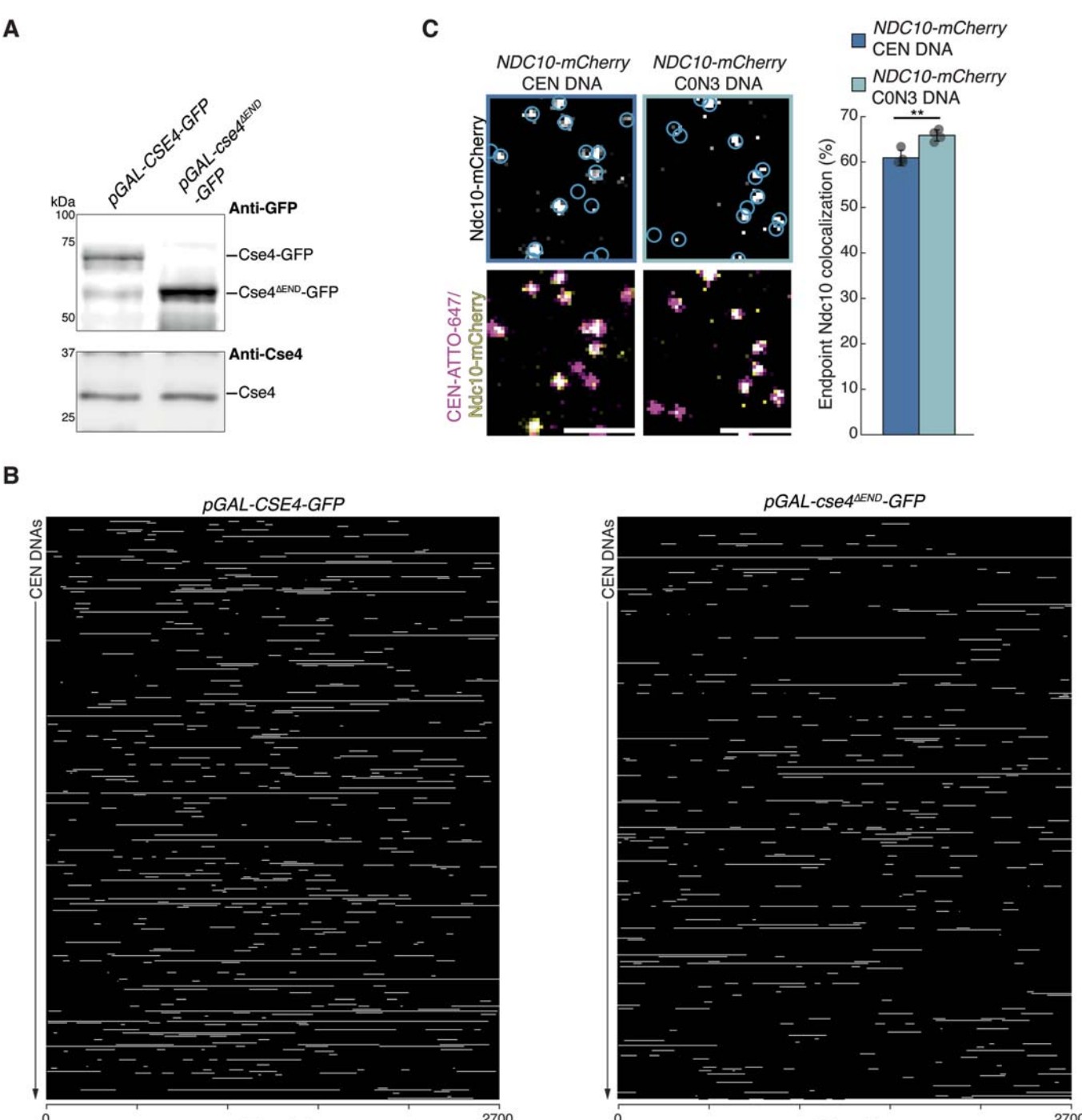

**Figure EV1. Cse4 recruitment is impaired by loss of the END region or non-native centromeric DNA composition.**

(**A**) Anti-GFP and anti-Cse4 immunoblots of whole-cell extract of *pGAL-CSE4-GFP* (left) and *pGAL-cse4^ΔEND^-GFP* (right). (**B**) Example time-lapse TIRFM colocalization assay plots of residences of Cse4 on CEN DNA per imaging sequence in *pGAL-CSE4-GFP* (SBY22273) extracts (left) or *pGAL-cse4^ΔEND^-GFP* (SBY22803) extracts (right). Each row represents one identified CEN DNA with all identified residences shown over entire imaging sequence (2700 s) for Cse4-GFP. (**C**) Example images of TIRFM endpoint colocalization assays. Top panels show visualized Ndc10-mCherry on CEN DNA (top-left panel), or on C0N3 DNA (top-right panel) in *NDC10-mCherry* (SBY8315) extracts with colocalization shown in relation to identified DNAs in blue circles. Bottom panels show overlay of CEN or C0N3 DNA channel (magenta) with Ndc10-mCherry (yellow), Scale bars 2 μm. Graph indicates quantification of Ndc10-mCherry endpoint colocalization with CEN DNA or C0N3 DNA (61 ± 1.7%, 66 ± 1.2%, avg ± s.d. $n = 4$ experiments, each examining ~1000 DNA molecules from different extracts, ** indicates significant difference with two-tailed $P$ value of 0.003).

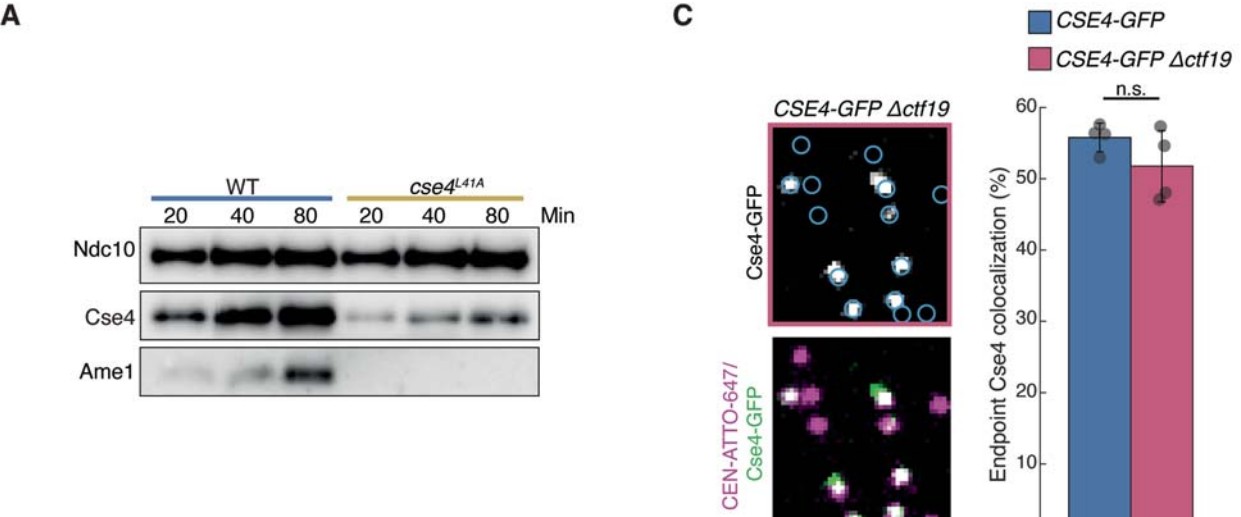

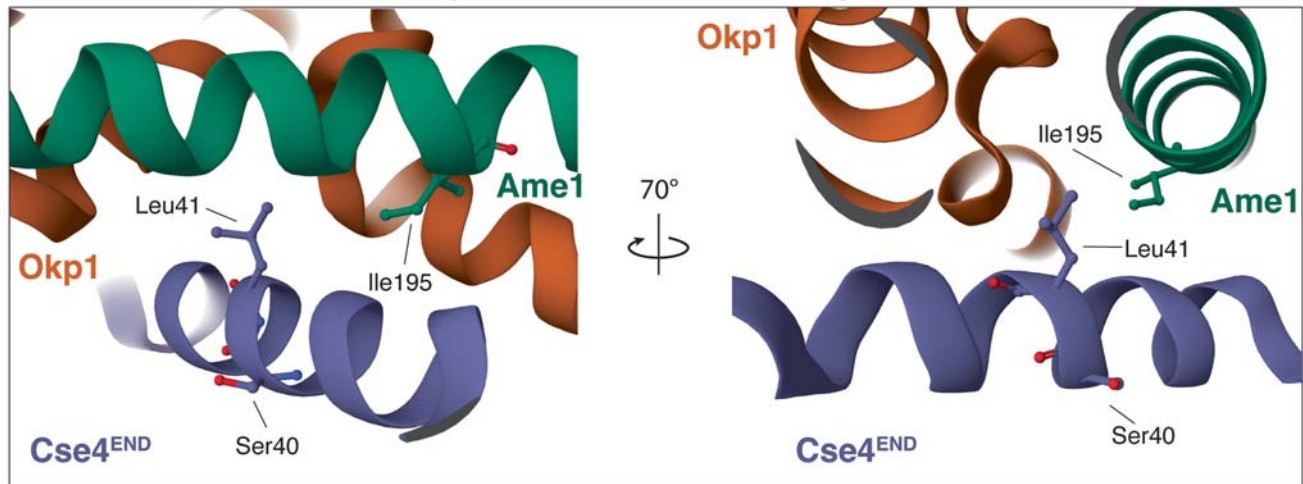

**Figure EV2.** Cse4 centromere recruitment is impaired by loss of the END binding to Okp1/Ame1 but not loss of the Ctf19 CCAN component.

(A) Immunoblots of bulk kinetochore assembly assays in WT (SBY3) (blue) or *cse4*$^{L41A}$ (SBY22466) (orange) extracts on CEN DNA. Centromere DNA-bound proteins were analyzed by immunoblotting with the indicated antibodies. (B) Structure of Cse4$^{END}$ (purple) in complex with Okp1/Ame1 (orange and green), highlighting hydrophobically packed residues Ile195 of Ame1 and Leu41 of Cse4 as well as Cse4 Ser40. Image of 8TOP adapted from (Deng et al, 2023) and created with Mol* (Sehnal et al, 2021). (C) Example images of TIRFM endpoint colocalization assays. Top panels show visualized Cse4-GFP on CEN DNA in *CSE4-GFP Δctf19* (SBY20038) extracts (top panel) with colocalization shown in relation to identified CEN DNA in blue circles. Bottom panels show overlay of CEN DNA channel (magenta) with Cse4-GFP (green), scale bars 2 μm. Graph indicates quantification of Cse4-GFP endpoint colocalization with CEN DNA in extracts from *CSE4-GFP* or *CSE4-GFP Δctf19* genetic backgrounds (56 ± 2.0%, 52 ± 5.0%, avg ± s.d. *n* = 4 experiments, each examining ~1000 DNA molecules from different extracts, n.s. indicates two-tailed *P* value of 0.2).

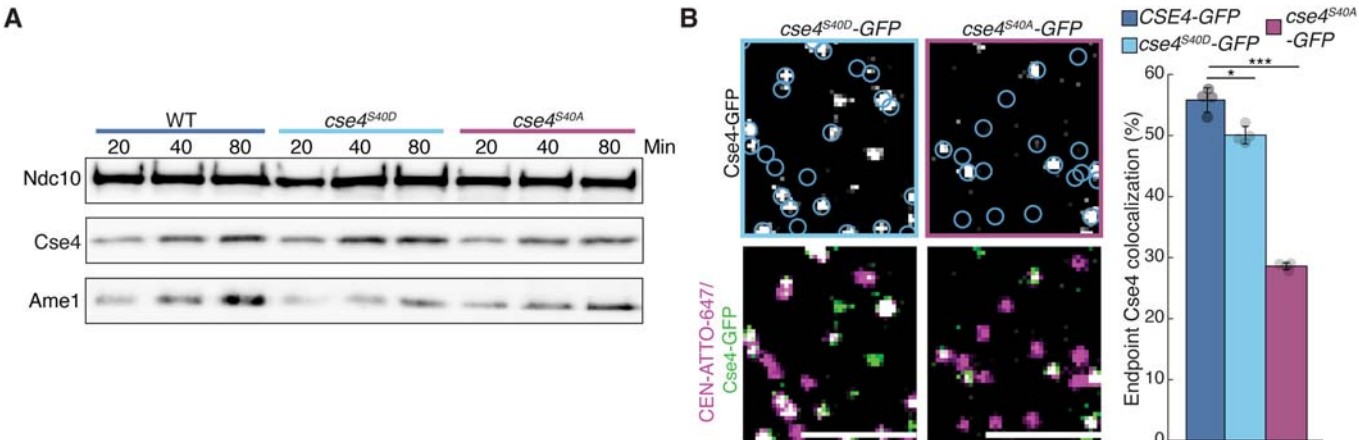

**Figure EV3. Phosphorylation of the Cse4 END at S40 regulates Cse4 levels during kinetochore assembly.**

(A) Bulk kinetochore assembly assays on centromeric DNA in WT (SBY4) (left), cse4$^{S40D}$ (SBY22401) (middle), or cse4$^{S40A}$ (SBY22405) (right) cell extracts. DNA-bound proteins were analyzed by immunoblotting with the indicated antibodies. (B) Example images of TIRFM endpoint colocalization assays. Top panels show visualized Cse4-GFP in extracts containing cse4$^{S40D}$-GFP (SBY20017) (top-left panel) or cse4$^{S40A}$-GFP (SBY20019) (top-right panel) on CEN DNA with colocalization shown in relation to identified CEN DNA in blue circles. Bottom panels show overlay of CEN DNA channel (magenta) with Cse4-GFP (green), Scale bars 3 μm. Graph indicates quantification of endpoint colocalization with CEN DNA of Cse4-GFP in extracts containing cse4$^{S40D}$-GFP or cse4$^{S40A}$-GFP (50 ± 1.4%, 29 ± 0.5% respectively, avg ± s.d. $n = 4$ experiments, each examining ~1000 DNA molecules from different extracts, * indicates significant difference between *CSE4-GFP* and cse4$^{S40D}$-GFP with two-tailed *P* value of.006, *** indicates significant difference between *CSE4-GFP* and cse4$^{S40A}$-GFP with two-tailed *P* value of 1.2E-4).

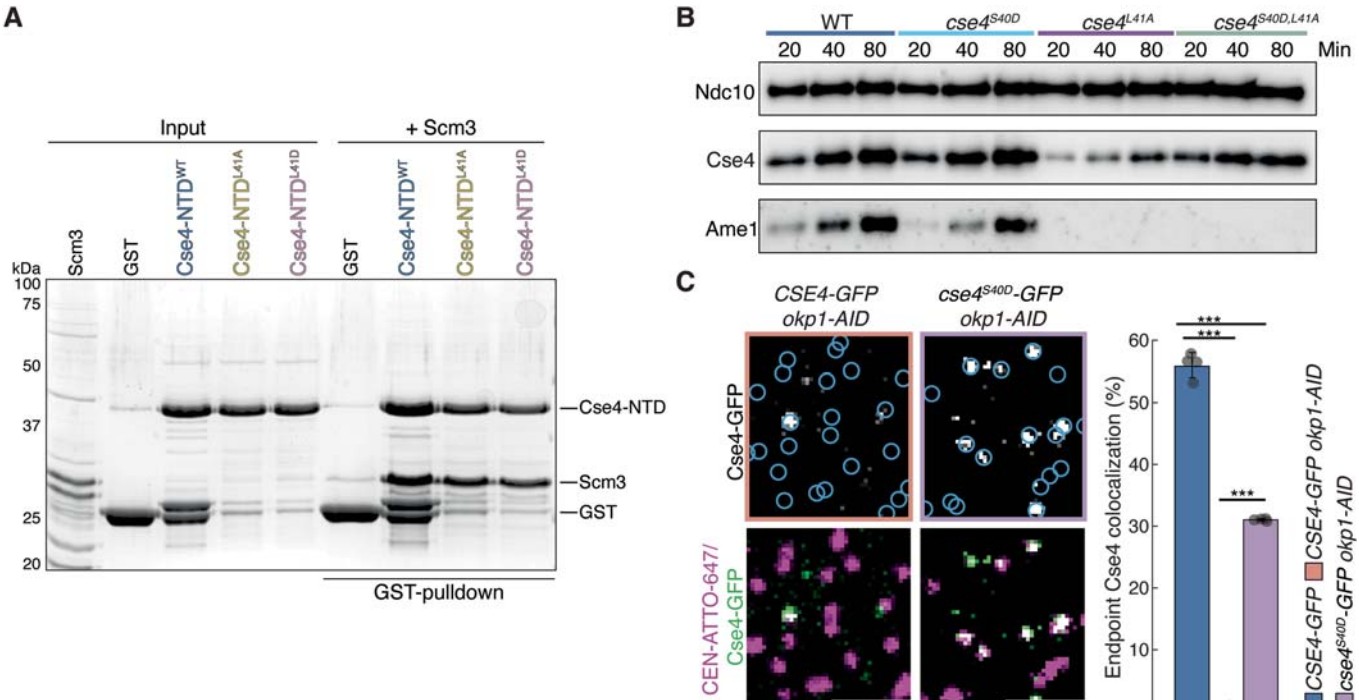

**Figure EV4. Reduced Cse4 recruitment to centromeric DNA due to disruption of Okp1/Ame1 binding to Cse4 END in *cse4^L41A* mutant or by depletion of *OKP1* is rescued by Cse4^S40D mutant.**

(**A**) SDS-PAGE of GST pull-down assays of immobilized GST-Cse4-NTD^WT, GST-Cse4-NTD^L41A and GST-Cse4-NTD^L41D to test binding of recombinant Scm3. (**B**) Bulk kinetochore assembly assays on CEN DNA in WT (SBY21863) extracts, *cse4^S40D* (SBY20017) extracts, *cse4^L41A* (SBY22811) extracts, or *cse4^S40D,L41A* (SBY22914) extracts. DNA-bound proteins were analyzed by immunoblotting with the indicated antibodies. (**C**) Example images of TIRFM endpoint colocalization assays. Top panels show visualized Cse4-GFP on CEN DNA in *CSE4-GFP okp1-AID* (SBY22987) extracts (top-left panel), or *cse4^S40D-GFP okp1-AID* (SBY20348) extracts (top-right panel) with colocalization shown in relation to identified CEN DNA in blue circles. Bottom panels show overlay of CEN DNA channel (magenta) with Cse4-GFP (green), scale bars 2 μm. Graph indicates quantification of Cse4-GFP endpoint colocalization with CEN DNA in extracts from *CSE4-GFP*, *CSE4-GFP okp1-AID*, or *cse4^S40D-GFP okp1-AID* genetic backgrounds (55 ± 0.5%, 1 ± 0.2%, or 31 ± 0.2%, avg ± s.d. $n = 4$ experiments, each examining ~1000 DNA molecules from different extracts, *** indicates significant difference between *CSE4-GFP* and *CSE4-GFP okp1-AID* with two-tailed $P$ value of 1.3E-5, between *CSE4-GFP* and *cse4^S40D-GFP okp1-AID* with two-tailed $P$ value of 1.5E-4, and between *cse4^S40D-GFP okp1-AID* and *CSE4-GFP okp1-AID* with two-tailed $P$ value of 5.4E-13).

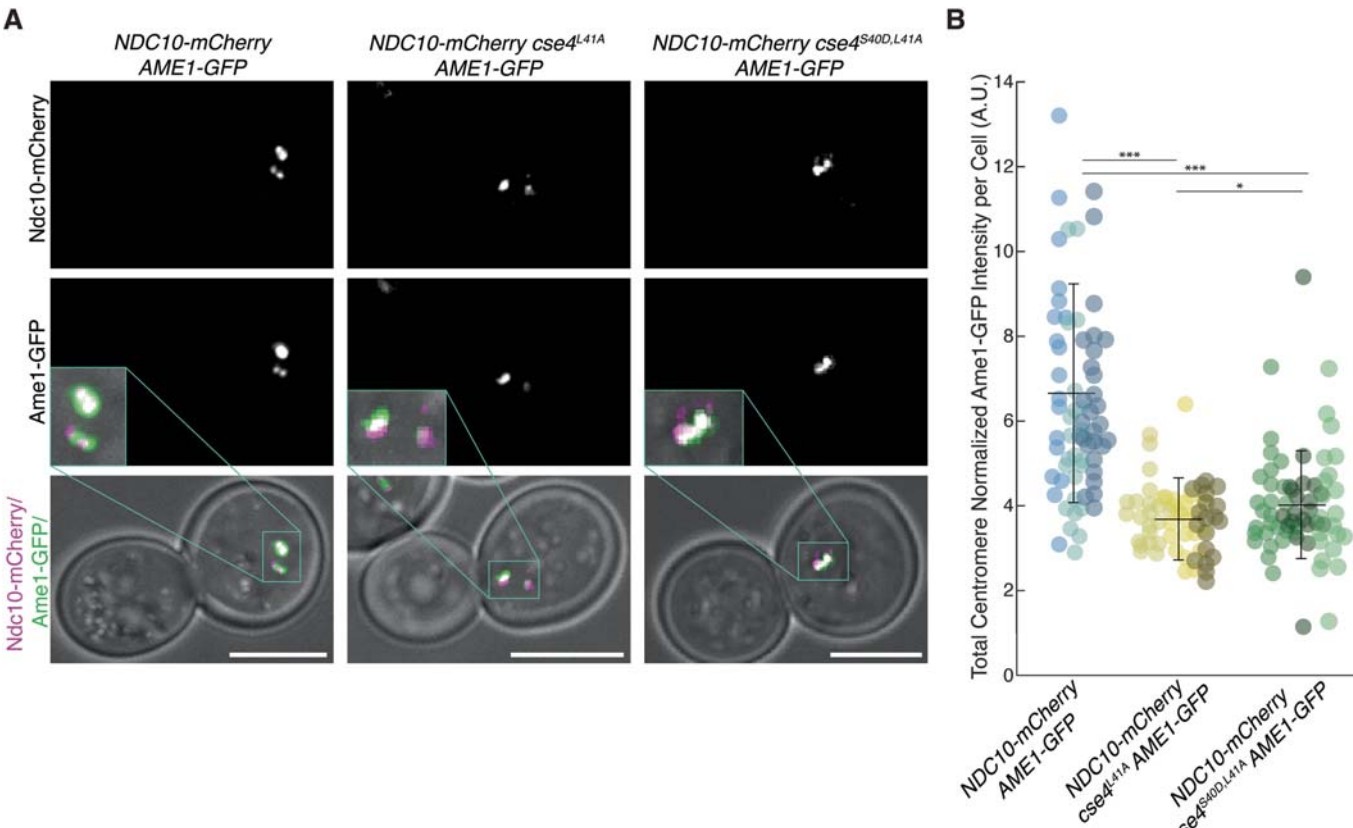

**Figure EV5.   Cse4^S40D,L41A mutant does not restore Okp1/Ame1 localization.**

(**A**) Example fluorescence microscopy images of *NDC10-mCherry AME1-GFP* (SBY23099 - left), *NDC10-mCherry cse4^L41A AME1-GFP* (SBY23295 - middle) and *NDC10-mCherry cse4^S40D,L41A AME1-GFP* (SBY23237 - right) cells showing visualized Ndc10-mCherry (top panels), Ame1-GFP (middle panels) and overlay of Ndc10-mCherry (magenta) and Ame1-GFP (green) on plane-polarized illumination of cell. Expanded region around kinetochores highlighted (middle panel inset). Scale bars 5 μm. (**B**) Graph indicates quantification of Ndc10-mCherry normalized centromere-associated Ame1-GFP intensity per cell of *NDC10-mCherry AME1-GFP* (left), *NDC10-mCherry cse4^L41A AME1-GFP* (middle) and *NDC10-mCherry cse4^S40D,L41A AME1-GFP* (right) cells (6.7 ± 2.6%, 3.7 ± 1.0%, 4.0 ± 1.3%, avg ± s.d. $n = 3$ experiments, each examining ~ 25 cells). * Indicates significant difference as determined by *t* test (*NDC10-mCherry AME1-GFP*: *cse4^L41A AME1-GFP* P value of 2.7E-14, *NDC10-mCherry AME1-GFP* : *NDC10-mCherry cse4^S40D,L41A AME1-GFP* P value of 8.6E-12 and *NDC10-mCherry cse4^L41A AME1-GFP* : *cse4^S40D,L41A NDC10-mCherry AME1-GFP* P value of.09). Each spot represents calculated intensity for one cell, different colors indicate biological replicates.

