## [Peer Review File · The EMBO Journal]

Stable centromere association of the yeast histone variant Cse4 requires its essential N-terminal domain

Andrew Popchock, Sabine Hedouin, Yizi Mao, Charles Asbury, Andrew Stergachis, and Sue Biggins

Corresponding author(s): Sue Biggins (sbiggins@fhcrc.org)

Review Timeline:

Submission Date:	23rd Jul 24
Editorial Decision:	26th Aug 24
Revision Received:	8th Nov 24
Editorial Decision:	29th Nov 24
Revision Received:	5th Dec 24
Accepted:	6th Dec 24

Editor: Hartmut Vodermaier

Transaction Report:

Dr. Sue Biggins
Fred Hutchinson Cancer Center
Division of Basic Sciences
1100 Fairview Ave N
Mailstop A2-168, PO B0x 19024
Seattle, WA 98109

26th Aug 2024

Re: EMBOJ-2024-118553
Stable centromere association of the yeast histone variant Cse4 requires its essential N-terminal domain

Dear Sue,

Thank you again for submitting your study on Cse4 nucleosome stabilization at centromeres to The EMBO Journal. It has now been assessed by three expert referees, whose reports you will find copied below. Since all reviewers appreciate the importance and interest of the work and are in principle supportive of its publication, we shall be happy to consider a revised manuscript further. As the reports do nevertheless raise several queries, I would encourage you to get back to me with tentative point-by-point response already during the early stages of revision, so that we might discuss how these open points could best be answered; in particular, referee 2's question about either strengthening or removing biochemical data would be good to discuss. As always, I would be happy to offer an extended revision period if this should be helpful in order to decisively address key referee concerns.

Detailed information on preparing, formatting and uploading a revised manuscript can be found below and in our Guide to Authors. Thank you again for the opportunity to consider this work for The EMBO Journal, and I look forward to hearing from you in due time.

With kind regards,

Hartmut

9) To facilitate reproducibility and cross-laboratory adoption of methodologies, please structure the Materials & Methods section as outlined in our guide to authors, including a completed Reagents and Tools Table that can be downloaded from our author guidelines as well (<https://www.embopress.org/page/journal/14602075/authorguide#structuredmethods>).

10) Digital image enhancement is acceptable practice, as long as it accurately represents the original data and conforms to community standards. If a figure has been subjected to significant electronic manipulation, this must be clearly noted in the figure legend and/or the 'Materials and Methods' section. The editors reserve the right to request original versions of figures and the original images that were used to assemble the figure. Finally, we generally encourage uploading of numerical as well as gel/blot image source data; for details see: embopress.org/page/journal/14602075/authorguide#sourcedata

At EMBO Press, we ask authors to provide source data for the main manuscript figures. Our source data coordinator will contact you to discuss which figure panels we would need source data for and will also provide you with helpful tips on how to upload and organize the files.

In the interest of ensuring the conceptual advance provided by the work, we recommend submitting a revision within 3 months (24th Nov 2024). Please discuss the revision progress ahead of this time with the editor if you require more time to complete the revisions. Use the link below to submit your revision:

Link Not Available

Referee #1:

Accurate chromosome segregation requires the assembly of a unique centromeric locus in many species. CENP-A/CSE4 containing centromeric nucleosome assembly and retention are critical features of a functional centromere. The current work focused on the genetically established budding yeast centromere in which a single Cse4 nucleosome is assembled into the CDEI/II/III DNA sequence. The extremely interesting aspect of this, which serves as the rationale for this manuscript, is that the sequences within CDEII are disfavored for nucleosome formation. This is shown by the investigators in a previous manuscript. The current manuscript from Popchoc et al. demonstrates role of the N-terminus of Cse4 in stabilizing the nucleosome at centromeric sequences. Overall, the work is an excellent dissection of the roles of the OA complex, the Ipl1 kinase and the scm3 chaperone in mediating the effects of the END domain on centromeric nucleosome stability. The manuscript includes several very quantitative and inventive in vitro assays to assess how the CSE4p N-terminus and its binding partners influence Cse4p stability. I have only minor comments on the manuscript to convey, as I think the manuscript is very thorough and provides a unique insight into the mechanisms that support centromere nucleosome stability.

1. The fiber-seq experiments for the *okp1-AID* and *cse4-AID* cell lines in figure 1 are great. It would be nice to see that the *cse4* END mutant shows a similar pattern of accessibility in vivo.

2. The endpoint experiments do not delineate between loading and stability. Is it possible that the Ipl1 mutations and the Cse4 mutations that alter Scm3 binding are affecting loading rather than stability? Can the authors demonstrate that Ipl and END mutants do not alter initial assembly?

3. What is the effect of the Ipl depletion in the context of *Ame1* mutation. Are these the same or different pathways?

Referee #2:

20240807

In this paper, Popchock et al. study protein interactions at the inner kinetochore. The major strength of this manuscript is that it uses two powerful techniques to bring totally new information to the table. The first technique, which is used only in the first figure, is a chromatin fiber mapping technology, which conclusively shows the DNA footprint of the centromeric nucleosome/CCAN. This is exciting to see. The second technique, which is used throughout the manuscript, merges an optimized in vitro kinetochore reconstitution strategy with TIRF microscopy, enabling the authors to visualize and measure individual kinetochores. The same authors reported this technique in a separate recent publication, so the assay itself is not totally new, but the sky is the limit for what this experiment and its variants can tell the field. More, please!

The main conclusion of the paper is that both Okp1-Ame1 and Scm3 stabilize the centromeric histone protein, Cse4, at centromeres. Ctf19 complex (Ame1-Okp1)-dependent Cse4 stabilization has been shown in meiotic prophase and is less surprising. Scm3-dependent Cse4 stabilization even after Cse4 deposition is somewhat surprising. The major strength of the paper is the ability to directly visualize and quantify these binding interactions in a near-native context, which is totally amazing. The major weakness of this paper is that the nature of the interaction between Scm3 and the Cse4 N-terminal region is not well-described. Previous work on this subject has not been definitive, and the biochemical assays done in the current manuscript are suggestive but leave substantial room for argument, especially regarding Cse4 phosphorylation and its function. There are clear genetic and functional data showing that Cse4-S40D can indeed rescue cells from inefficient Ame1-Okp1-Cse4 contact, which is new and interesting.

My recommendation is that the authors rework the paper to remove most of the biochemical Scm3-Cse4 binding data/claims and focus instead on the phenomena that are clearly supported by the data: Ame1 and Scm3 both are required for stable Cse4-CEN association. Cse4-S40D can suppress Ame1-Okp1 binding defects. Ipl1 enhances Cse4 loading.

Below, I pose specific questions. I do not think the authors need to include quantitative biochemical assays to publish this work, but I think they essentially have a choice between expanding the binding assays to make them quantitative or removing several of the current pulldown-style experiments and publishing a paper that focuses much more on the phenomenon, substantiated by genetic data, rather than a direct binding interaction. I am strongly in favor of publishing this paper after addressing these concerns.

Major questions:

- The idea that Cse4 phosphorylation controls differential affinity for Scm3 and Ame1/Okp1 is plausible, but the current data is not convincing on this point.
 - o The pulldown data showing differential binding of phospho-mimetic Cse4-N (S40D) to Scm3 is not rigorous, since the different ratios used are not meaningful in a biochemical sense.
 - o To determine whether S40D enhances binding affinity, a quantitative assay like fluorescence polarization or ITC should be done (see comments above).
- The cited published data for an Scm3-Cse4-N interaction is not definitive. The authors of the cited paper suggest the following: "We have established that both proteins are intrinsically disordered and form a protein complex in vitro with no significant gain in structure." Is this the view of the authors of the current paper? I'm struggling to understand how I should think about the proposed Scm3-Cse4-N interaction. Adding to the confusion, the current manuscript shows that Cse4-N delta-END does not bind Scm3 in a pulldown assay. Since Cse4-END is helical when bound to Ame1-Okp1 and in all predictions, this seems at odds with the published Scm3-Cse4 binding data.
 - o The cited binding data also suggests Scm3 binds AO. Do the authors think this is relevant? And has this been tested?
 - o Alpha fold 3 predictions show no meaningful interaction between Scm3 and Cse4-N (unmodified or phospho-S22, -S30, -S40). This is very different from the AO-Cse4-N predictions.
- The diagram at the end is not fully supported by the data (see above). Here are some specific questions: i) why isn't CBF3/Ndc10 preloaded on the CEN DNA? ii) Why does Scm3 appear to fall off in steps 4 and 5? iii) Isn't Scm3 recruited through CBF3? Is there a reason to think this interaction would not persist (allowing for exchange, which is known)? In my view, a more conceptualized schematic showing mainly the studied factors (CEN, Scm3, Cse4, and AO) and their observed relationships/dependencies in the reconstitution assay would be more helpful here. I don't think the protein-suggestive cartoons are necessary, since they give the impression of a worked out structural mechanism, but that is not the case.
- The S40 site is reported as an Ipl1 site, but the published data supporting this idea is not conclusive. Are there examples of known R/K - X - X - S/T - L/I Ipl1 sites? The data showing Ipl1-dependent Cse4 stabilization and compensation by S40D hold together, but the site itself and the relatively weak underlying data is an uncomfortable fact. Are there other potential substrates that may explain the Ipl1-AID and S40D rescue results? MIND, which might stabilize everything, is probably not recruited at high enough efficiency (fractional occupancy). What about Mif2? Ndc10? Both are Ipl1 substrates, of course.
- The authors propose that Ipl1 proximity serves a licensing function for Cse4 nucleosome assembly. It isn't clear why kinase activity should be required. Wouldn't CBF3 binding (for instance) be sufficient? Kinase activity might provide a temporal cue, but there is no obvious reason to need this, since the main assembly event in yeast is thought to be immediately after passage of the replication fork.
- Given their hypothesis about Cse4 licensing, do the authors think that the C0N3 loading is poor because the DNA gets loaded with normal (dark) H3 nucleosomes?

Minor questions:

- There seems to be an extra protected region to the right of CEN5 in Figure 1B. Is this the R-CDEIII assembly site for a second CBF3? Has this been observed before? See PMID30478265, Figure 4.
- The image segmentation in figure 2C is questionable. Was the segmentation threshold manually defined? Could this be improved?
- The Cse4 internal GFP tag is not mentioned in the main text. I still see some papers using C-terminal Cse4 tags, and it would be good to emphasize the importance of the internal tag in a single short sentence/phrase somewhere.
- The blot in Figure 4A looks like it was cut at an unlucky spot. Can this be rerun? Ideally, this and the blot in figure 4C would be done on the same membrane and in the same experiment so that the cse4-S40D mutant can be directly compared with WT.
- The experiment showing co-binding to Cse4-N by AO and Scm3 is not rigorous. There seems to be excess Cse4-N in this experiment, so it's not clear we should expect any competition in the first place, even with perfectly competitive binders.
- It looks like the Cse4 off rates do not match Popchock, 2023. For example, Figure 2D in 2023 vs. Figure 3E in the current manuscript. Can the authors explain this? If these are different lysates, is there a good way to quantify the lysate-specific activity? What parameters vary between lysates (off rate? Final loading amount? Etc.) and does this correlate with concentration, cell health, etc?
- Figure 6B and C look like they use the same AME1 WT data. Shouldn't these just be shown in a single plot? In other words, two plots for the figure, one for Cse4 and one for Ame1 localization? Looking more closely, this also seems true of Figure 4B-D (IPL1, and ipl1-AID). Why not just merge them?
- In many places, the authors reference structures of the assembled kinetochore, but (as it is rightly acknowledged, though not firmly enough) the referenced cryo-EM structures probably do not reflect the physiologic assemblies, so it is strange to reference them on the one hand as models for what is assembling in extract while on the other hand present and point to previous data (Popchock, 2023, for instance, and in the cited papers themselves) that indicates these nucleosome-containing structures are incorrect.
 - o "...sequence used in the most recent and complete structural studies..." is a great example. What is meant by "complete" here? There seems to be a logical disconnect.
- Hinshaw and Harrison (2019) reported Ame1-Okp1 binding to Cse4 in a pulldown assay and showed how this likely fits into the overall kinetochore structure at the same time or before this was reported by the papers cited on this point, so this is an odd omission.
- Phosphorylation-dependent inner kinetochore stabilization has recently been described in yeast (Klemm et al, 2023, Hinshaw et al, 2023, Hagemann et al, 2022) and vertebrates (Watanabe et al, 2019, Ariyoshi et al, 2021). CENP-C and Mif2 are Ipl1 targets, though the functions of these sites are not known. Can the authors rule out Mif2/CENP-C phosphorylation-dependent stabilization as the reason for the Ipl1-AID results?
- It would be great to know whether Cse4-END binds AO in the assembled kinetochore or if the mutations that disrupt this interaction just prevent something stable from assembling. Can this be inferred from the data somehow? It would help clarify whether the absence of Cse4-AO contact in cryo-EM reconstructions means the biochemical reconstitutions are missing something.
 - o "Although the NDT is essential, it is absent..." - Since Cse4-END is structured when bound to AO, it's not quite right to say that Cse4-N disorder is what prevents us from seeing it in the cryo-EM structures.
- Some of the figure titles, especially in the supplement, seem to be incorrect. Titles for figures S1 and S2 should be updated.
- Figure S7B and other imaging experiments: were these cultures staged or asynchronous?
- Page 14: cse4-S40,L41A the Cse4 mutation is incomplete
- Page 16: "...so we presume [Scm3] initially targets Cse4 to centromeres via [Cse4 histone fold]..." but we don't know whether these interactions are mutually exclusive.

Referee #3:

Title: Stable centromere association of the yeast histone variant Cse4 requires its essential N-terminal domain.

Authors: Popchock et al.

Popchock et al., describe a new molecular mechanism for Cse4 stabilization at centromeres using elegant technique using single molecule fluorescence assays to monitor Cse4 during kinetochore assembly. These studies define a novel role for the essential N-terminus domain (END) of Cse4 in kinetochore assembly. More specifically they describe a role for phosphorylation of S40 in the END domain previously shown to be a substrate for Ipl1 by Boeckmann et al., 2013. They determined that Ipl1-mediated phosphorylation of Cse4-S40 plays a role in Scm3-mediated Cse4 stabilization at the kinetochore. Furthermore, the results show that interaction of essential components namey Okp1/Ame1 (OA) of the COMA complex and the interaction of Scm3 with END act independently to stabilize Cse4 at centromeric chromatin. A phosphomimetic mutant of Cse4 (Cse4 40SD) in the Cse4 END enhances Scm3 binding and can restore Cse4 recruitment in mutants defective in OA binding. The research objectives of this study are logical, and experimental approaches are robust and technically sound. The data presented in the manuscript are thorough, statistically methods used for data analysis are accurate, and data provides molecular insights into the role of Cse4-END in kinetochore function and chromosome segregation and should be of broad interest to the chromosome biology community. Listed below are comments.

Major Comments:

1. OA has not been defined in the abstract. Please do so [EDITOR's NOTE: please avoid the unnecessary abbreviation OA throughout the main manuscript text, and rather stick with Okp1/Ame1 for clarity]. Authors have shown in Figure 1B that depletion of OA affects the Cse4 nucleosome occupancy at the kinetochores. It is however unclear whether the phenotype observed here is unique to Cse4 or other inner kinetochore proteins are also affected. Given the extensive studies with human cells on the interaction of CENP-A and CENP-C (Mif2 homolog in budding yeast) it is of interest to examine the levels of at least one more inner kinetochore protein for example, Mif2 upon OA depletion, which will give additional support to their observation and provide a deeper mechanistic insight into the role of OA in kinetochore assembly.
2. Authors have conducted experiments to examine the role of END domain of Cse4 and defined its novel function in the kinetochore assembly regulated by the phosphorylation of Cse4-S40. However, Cse4 contains S33 in the END that is also phosphorylated (Boeckmann et al. 2013 MBoC). A role for S33 phosphorylation in centromeric deposition of Cse4 was reported by Hoffman et al., 2018. The authors should examine the role of S33 as well, double mutants and single mutants for both residues. What is the phenotype of S40A or S33A in their assays? For example, in Figure 7A, the effect of cse4 S40A or S33A on growth of L41A.
3. How do they know S40D just causes a structural change rather than being phosphomimetic? Why not use S40E?
4. Authors have concluded that Scm3 interacts with Cse4-CATD for the recruitment of Cse4 at the kinetochores. Once Cse4 nucleosome assembles on the CEN DNA, Scm3 dissociates from Cse4-CATD and then interacts with the Cse4-END. This implies that Scm3 and Cse4-END interactions most likely may be occurring outside of the S-phase. It is important to address the cell cycle dependent roles for the interaction of OA and Scm3 with END domain in stabilization of Cse4 at centromeric chromatin using extracts from synchronized cells. Along the same lines, when does S40 get phosphorylated by Ipl and whether this helps to address the differential roles of Ipl1-mediated phosphorylation in error correction during mitosis versus stabilization of Cse4? Extracts from synchronized cells should help them address possible differential role of Ipl1 mediated phosphorylation, Scm3, OA interact with END domain and provide a better explanation for their results.
5. Some observations from previous studies cannot be explained by the current study. This presents a conundrum for this reviewer. For example, the N-terminal domain of Cse4 (lacking 129 amino acids) containing END is dispensable for haploid growth when overexpressed (Morey et al. 2004, Eukaryotic Cell). The authors have mentioned this data without further discussion of how their data can explain these observations. The cells are alive even without END... how does one explain the role of OA and Scm3 in these cells that lack the N-terminal domain? Can the C-terminus of Cse4 recruit Ame1 and Okp1. Along the same lines, overexpression of Cse4 can suppress scm3 depleted cells. It is important for authors to include a discussion on these observations to reconcile their data.
6. Figure 7 shows growth defects of cse4-mutants at high temperature and a correlation with the de-clustering of Cse4-GFP foci at the kinetochores. It is of interest to examine whether these mutants exhibit chromosome segregation defects?
7. Model (Fig 8) describes the dual function of Scm3 via Cse4 CATD and NTD domains. Is there any evidence that CATD-bound Scm3 is released when DNA wraps to form nucleosome? Is this a relocation?

Other comments:

1. Why not also examine assembly of another COMA component, Ctf19 to the centromere in the context of END, and S40D. Unlike Ame1 and Okp1, Ctf19 is not essential, so extracts from a ctf19 Δ strain will address the role of Ame1 and Okp1 is related to stabilizing Cse4 at centromere or some other role?
2. If phosphorylation of S40 is essential for stabilization of Cse4 at CEN, we may expect the Cse4 4SA will cause defects in chromosome segregation but based on Boeckmann et al., 2013 the Cse4 4SA does not exhibit defects in chromosome segregation. How do the authors reconcile the role of phosphorylation based on these observations?
3. Lethality of Scm3 depletion can be rescued by OE Cse4, I am wondering if under this condition whether OA associate with the centromere? Will this phenotype be altered upon deletion of CAF1?
4. Curious why authors do GST pull-down instead of in vivo CoIP?
5. Previous studies have shown that Cdc5 (Mishra et al., 2019) can also phosphorylate S40, have the authors examined the role of Cdc5 mediated phosphorylation of S40 in interaction with Scm3?
6. They have demonstrated a novel interaction of Cse4-END domain with its chaperone Scm3. Previous studies have shown that CATD of Cse4 interacts in vivo and in vitro with the HR-domain of Scm3 (Stoler et al. 2007 PNAS; Mizuguchi et al. 2007, Cell). It will be of great interest to define the domain within Scm3 that interacts with the Cse4-END domain.

We are grateful to the reviewers and editor for their thoughtful suggestions on how to improve the manuscript. We have made the following revisions in response to the comments:

Referee #1:

Accurate chromosome segregation requires the assembly of a unique centromeric locus in many species. CENP-A/CSE4 containing centromeric nucleosome assembly and retention are critical features of a functional centromere. The current work focused on the genetically established budding yeast centromere in which a single Cse4 nucleosome is assembled into the CDEI/II/III DNA sequence. The extremely interesting aspect of this, which serves as the rationale for this manuscript, is that the sequences within CDEII are disfavored for nucleosome formation. This is shown by the investigators in a previous manuscript. The current manuscript from Popchok et al. demonstrates role of the N-terminus of Cse4 in stabilizing the nucleosome at centromeric sequences. Overall, the work is an excellent dissection of the roles of the OA complex, the Ipl1 kinase and the scm3 chaperone in mediating the effects of the END domain on centromeric nucleosome stability. The manuscript includes several very quantitative and inventive *in vitro* assays to assess how the CSE4p N-terminus and its binding partners influence Cse4p stability. I have only minor comments on the manuscript to convey, as I think the manuscript is very thorough and provides a unique insight into the mechanisms that support centromere nucleosome stability.

1. The fiber-seq experiments for the *okp1-AID* and *cse4-AID* cell lines in figure 1 are great. It would be nice to see that the *cse4* END mutant shows a similar pattern of accessibility *in vivo*.

We agree this experiment would be strong evidence that the reason for the instability of Cse4 in the *okp1-AID* strain is directly mediated through its binding to the END domain of Cse4. However, this is a technically challenging experiment. We cannot construct a conditional *cse4ΔEND* construct because overexpression of the Cse4ΔEND protein has been reported to suppress a *cse4* deletion and therefore bypass the functions of the N-terminal domain. To perform a fiber-seq experiment, we will therefore need to construct a strain where the Cse4ΔEND protein is expressed from its endogenous promoter in the presence of a conditional endogenous Cse4-AID protein that we can degrade. While this is possible, it will also require that the mutant Cse4ΔEND protein is expressed at normal levels, something we will not know until we construct the strain. It is also unclear whether dimerization of the mutant with the endogenous Cse4-AID protein will lead to lower levels of the Cse4ΔEND protein when we add auxin to promote degradation of the Cse4-AID protein. Because of these challenges and the length of time it takes to perform the fiber-seq experiments, we did not attempt this experiment after discussing the challenges with the editor. While this leaves open the question of whether all of the effects of the *okp1-aid* mutant on Cse4 localization are through binding to the END domain, we have performed fiber-seq on other CCAN mutants that do not bind to Cse4 and they do not show a similar effect on Cse4 localization (data

added to Expanded View Figure 2C). It is therefore likely that OA binding to the N-terminus is a major contributor to the fiber-seq result in the manuscript.

2. The endpoint experiments do not delineate between loading and stability. Is it possible that the *Ipl1* mutations and the *Cse4* mutations that alter *Scm3* binding are affecting loading rather than stability? Can the authors demonstrate that *Ipl1* and *END* mutants do not alter initial assembly?

We have done time lapse imaging on the *Cse4-S40A* and *Cse4-S40D* lysates and found that they have a similar number of binding events. Therefore, they have similar arrival/loading behavior. We have not performed this on the *ipl1-AID* strain, but the most severe mutant is the *Cse4-END* mutant that also displays binding events and suggests that *Cse4* loading is not the primary defect in any of these mutants. We have provided supplementary example binding traces to support this conclusion (Expanded View Figure 1B) .

3. What is the effect of the *Ipl1* depletion in the context of *Ame1* mutation. Are these the same or different pathways?

We constructed an *ipl1-AID ame1-Y195* double mutant strain with *Cse4-GFP* and performed a *Cse4* TIRFM endpoint assay to address this question. In contrast to our expectation that they might have an even more severe defect in *Cse4* localization, we found that the double mutant behaved similarly to an *ipl1-AID* single mutant and had a 40% final *Cse4* co-localization with CEN DNA. This is higher than the *Cse4* localization in the *ame1-Y195* single mutant (17%). It is unclear why the *ipl1-AID* mutant suppresses the defect in *ame1-Y195* and it doesn't allow us to conclude how many pathways are potentially affected. Therefore, we therefore did not add this result to the revision but are happy to put it in if the reviewer thinks it is an important result.

Referee #2:

20240807

In this paper, Popchock et al. study protein interactions at the inner kinetochore. The major strength of this manuscript is that it uses two powerful techniques to bring totally new information to the table. The first technique, which is used only in the first figure, is a chromatin fiber mapping technology, which conclusively shows the DNA footprint of the centromeric nucleosome/CCAN. This is exciting to see. The second technique, which is used throughout the manuscript, merges an optimized in vitro kinetochore reconstitution strategy with TIRF microscopy, enabling the authors to visualize and measure individual kinetochores. The same authors reported this technique in a separate recent publication, so the assay itself is not totally new, but the sky is the limit for what this experiment and its variants can tell the field. More, please!

The main conclusion of the paper is that both *Okp1-Ame1* and *Scm3* stabilize the centromeric histone protein, *Cse4*, at centromeres. *Ctf19* complex (*Ame1-Okp1*)-dependent *Cse4* stabilization has been shown in meiotic prophase and is less

surprising. Scm3-dependent Cse4 stabilization even after Cse4 deposition is somewhat surprising. The major strength of the paper is the ability to directly visualize and quantify these binding interactions in a near-native context, which is totally amazing.

The major weakness of this paper is that the nature of the interaction between Scm3 and the Cse4 N-terminal region is not well-described. Previous work on this subject has not been definitive, and the biochemical assays done in the current manuscript are suggestive but leave substantial room for argument, especially regarding Cse4 phosphorylation and its function. There are clear genetic and functional data showing that Cse4-S40D can indeed rescue cells from inefficient Ame1-Okp1-Cse4 contact, which is new and interesting.

My recommendation is that the authors rework the paper to remove most of the biochemical Scm3-Cse4 binding data/claims and focus instead on the phenomena that are clearly supported by the data: Ame1 and Scm3 both are required for stable Cse4-CEN association. Cse4-S40D can suppress Ame1-Okp1 binding defects. Ipl1 enhances Cse4 loading.

Below, I pose specific questions. I do not think the authors need to include quantitative biochemical assays to publish this work, but I think they essentially have a choice between expanding the binding assays to make them quantitative or removing several of the current pulldown-style experiments and publishing a paper that focuses much more on the phenomenon, substantiated by genetic data, rather than a direct binding interaction. I am strongly in favor of publishing this paper after addressing these concerns.

Major questions:

- The idea that Cse4 phosphorylation controls differential affinity for Scm3 and Ame1/Okp1 is plausible, but the current data is not convincing on this point.
 - o The pulldown data showing differential binding of phospho-mimetic Cse4-N (S40D) to Scm3 is not rigorous, since the different ratios used are not meaningful in a biochemical sense.
 - o To determine whether S40D enhances binding affinity, a quantitative assay like fluorescence polarization or ITC should be done (see comments above).

To address the concerns of this reviewer, we have moved some of the biochemistry data to the appendix and somewhat downplayed the conclusion that the phosphorylation directly controls Scm3 binding throughout the paper. We have tried to focus on Ipl1 phosphorylation of the END domain instead where possible instead of saying Scm3 binding. However, the TIRF data does show an enhanced stability of Scm3 at centromeres when Cse4 is phosphorylated on S40, so there is some clear effect of this phosphorylation event on Scm3 localization at the centromere through the Cse4 N-terminus. In addition, we have also recapitulated the published data showing the binding of Scm3 to the END domain, so we think it is reasonable to state that Scm3 binds to the END domain.

- The cited published data for an Scm3-Cse4-N interaction is not definitive. The authors of the cited paper suggest the following: "We have established that both proteins are intrinsically disordered and form a protein complex in vitro with no significant gain in

structure." Is this the view of the authors of the current paper? I'm struggling to understand how I should think about the proposed Scm3-Cse4-N interaction. Adding to the confusion, the current manuscript shows that Cse4-N delta-END does not bind Scm3 in a pulldown assay. Since Cse4-END is helical when bound to Ame1-Okp1 and in all predictions, this seems at odds with the published Scm3-Cse4 binding data.

o The cited binding data also suggests Scm3 binds AO. Do the authors think this is relevant? And has this been tested?

o Alpha fold 3 predictions show no meaningful interaction between Scm3 and Cse4-N (unmodified or phospho-S22, -S30, -S40). This is very different from the AO-Cse4-N predictions.

We did not mean to imply that Scm3 binding to the END domain would have the same effect as OA binding to this domain. While we understand that this reviewer is skeptical about the interaction, we have confirmed the published NMR data (doi.org/10.1016/j.str.2024.03.002) with biochemical pulldown assays so we don't think it should be entirely ignored despite alpha fold 3 not predicting how they interact. As brought up in other comments, it is entirely possible Scm3 has many additional interactions with other kinetochore proteins and the "fuzzy" interaction between Cse4 and Scm3 is just one of many. We have therefore decided to leave some of the Scm3 binding in the manuscript but have downplayed its importance. We agree there are structural questions about this interaction that need to be tackled in the future.

- The diagram at the end is not fully supported by the data (see above). Here are some specific questions: i) why isn't CBF3/Ndc10 preloaded on the CEN DNA? ii) Why does Scm3 appear to fall off in steps 4 and 5? iii) Isn't Scm3 recruited through CBF3? Is there a reason to think this interaction would not persist (allowing for exchange, which is known)? In my view, a more conceptualized schematic showing mainly the studied factors (CEN, Scm3, Cse4, and AO) and their observed relationships/dependencies in the reconstitution assay would be more helpful here. I don't think the protein-suggestive cartoons are necessary, since they give the impression of a worked out structural mechanism, but that is not the case.

We appreciate the suggestion to focus the model and did not mean to imply more than we discovered. We have simplified the model (Figure 8) in the revision to focus only on the observations made in the manuscript and therefore left out other factors such as CBF3.

- The S40 site is reported as an Ipl1 site, but the published data supporting this idea is not conclusive. Are there examples of known R/K - X - X - S/T - L/I Ipl1 sites? The data showing Ipl1-dependent Cse4 stabilization and compensation by S40D hold together, but the site itself and the relatively weak underlying data is an uncomfortable fact. Are there other potential substrates that may explain the Ipl1-AID and S40D rescue results? MIND, which might stabilize everything, is probably not recruited at high enough efficiency (fractional occupancy). What about Mif2? Ndc10? Both are Ipl1 substrates, of course.

It was previously published that Cse4-S40 is an Ipl1 site (doi: [10.1091/mbc.E12-12-0893](https://doi.org/10.1091/mbc.E12-12-0893)). While S40 does not meet the common consensus site criteria, there are many examples of Ipl1 sites that have a slightly different consensus (for example, Mad3 **Ser10**: YSKKRISYMP (doi: [10.1101/gad.431507](https://doi.org/10.1101/gad.431507)), Mif2 **Ser277**: RTDSIIDR (DOI: [10.1016/j.cub.2023.01.012](https://doi.org/10.1016/j.cub.2023.01.012)). However, we agree that there are many additional Ipl1 targets at the kinetochore, including Mif2 and Ndc10, and we are interested in exploring their roles in Cse4 stability in the future. We have therefore made it clear in the revised manuscript that there may be one or more additional Ipl1 substrates involved in Cse4 stability and kinetochore assembly (Discussion, p. 18).

- The authors propose that Ipl1 proximity serves a licensing function for Cse4 nucleosome assembly. It isn't clear why kinase activity should be required. Wouldn't CBF3 binding (for instance) be sufficient? Kinase activity might provide a temporal cue, but there is no obvious reason to need this, since the main assembly event in yeast is thought to be immediately after passage of the replication fork.

We agree that Ipl1 is not a master switch for Cse4 assembly. We propose it helps to license because it would be yet another layer of control at the kinetochore to stabilize the nucleosome at the proper site. CBF3 is indeed a much stronger control over Cse4 deposition but the stability seems to be controlled by other kinetochore proteins, such as Okp1, Ame1 and Scm3. We propose that these many controls are part of a licensing mechanism to stabilize it at the centromere and adding phosphoregulation as another regulatory event helps to ensure stability at the proper locus. We did not mean to imply it would be strictly required.

- Given their hypothesis about Cse4 licensing, do the authors think that the C0N3 loading is poor because the DNA gets loaded with normal (dark) H3 nucleosomes?

This is a really interesting question. In our prior work (DOI: [10.1101/2023.01.20.524981](https://doi.org/10.1101/2023.01.20.524981)), we asked whether H3 assembled onto a very similar template and did not detect significant H3 assembly. Because this template had the Widom DNA that is in the C0N3 template, we don't believe the C0N3 template would be different and we have added this point to the text (p. 7).

Minor questions:

- There seems to be an extra protected region to the right of CEN5 in Figure 1B. Is this the R-CDEIII assembly site for a second CBF3? Has this been observed before? See PMID30478265, Figure 4.

There is indeed ~30bp of protected DNA downstream of the CDEIII element. While we don't know the nature of the proteins protecting this region, it agrees with recent structural data (DOI: [10.1126/sciadv.adg748](https://doi.org/10.1126/sciadv.adg748)) showing that CCAN extends ~30bp beyond CDEIII. However, as pointed out, it could potentially be a second CBF3 binding site. This extra protection has been observed before (<https://doi.org/10.1073/pnas.1104978108>). We will discuss this in our upcoming manuscript where we publish the full genomic analysis of the fiber-seq data. We did not

include it here since it didn't seem relevant to the current manuscript focus but appreciate the reviewer noticing this interesting point.

- The image segmentation in figure 2C is questionable. Was the segmentation threshold manually defined? Could this be improved?

We did not realize this image appeared concerning. It was not handled differently from any other images but we replaced it to avoid concerns.

- The Cse4 internal GFP tag is not mentioned in the main text. I still see some papers using C-terminal Cse4 tags, and it would be good to emphasize the importance of the internal tag in a single short sentence/phrase somewhere.

We appreciate this suggestion and have made it clear we are using an internal tag (p. 7).

- The blot in Figure 4A looks like it was cut at an unlucky spot. Can this be rerun? Ideally, this and the blot in figure 4C would be done on the same membrane and in the same experiment so that the cse4-S40D mutant can be directly compared with WT.

Since these blots are not quantitative to begin with, we moved them to the appendix (Appendix Figure S1B and S1C) and left the quantitative TIRF results in the main manuscript.

- The experiment showing co-binding to Cse4-N by AO and Scm3 is not rigorous. There seems to be excess Cse4-N in this experiment, so it's not clear we should expect any competition in the first place, even with perfectly competitive binders.

We have removed this experiment given the reviewer's concern since it was not important to the conclusions of our manuscript.

- It looks like the Cse4 off rates do not match Popchock, 2023. For example, Figure 2D in 2023 vs. Figure 3E in the current manuscript. Can the authors explain this? If these are different lysates, is there a good way to quantify the lysate-specific activity? What parameters vary between lysates (off rate? Final loading amount? Etc.) and does this correlate with concentration, cell health, etc?

The reported off rates are not analyzing Cse4 alone so they cannot be compared. Instead, the off-rate analysis is measuring a ternary complex of two proteins with the centromeric DNA. In this manuscript, we analyzed the off rate of Cse4 with either Scm3 or Ame1. In the previous paper, we analyzed Cse4 with Ndc10 (Figure 3E, 2023).

- Figure 6B and C look like they use the same AME1 WT data. Shouldn't these just be shown in a single plot? In other words, two plots for the figure, one for Cse4 and one for Ame1 localization? Looking more closely, this also seems true of Figure 4B-D (IPL1, and ipl1-AID). Why not just merge them?

We merged the graphs in Figure 4B as requested. In Figure 6, there are two different proteins being assayed (Cse4 and Ame1) so we need to keep separate graphs for each. In addition, there are various mutants for each protein, so we kept these graphs separate for clarity.

- In many places, the authors reference structures of the assembled kinetochore, but (as it is rightly acknowledged, though not firmly enough) the referenced cryo-EM structures probably do not reflect the physiologic assemblies, so it is strange to reference them on the one hand as models for what is assembling in extract while on the other hand present and point to previous data (Popchock, 2023, for instance, and in the cited papers themselves) that indicates these nucleosome-containing structures are incorrect.

o "...sequence used in the most recent and complete structural studies..." is a great example. What is meant by "complete" here? There seems to be a logical disconnect.

Our use of the word "complete" referred to the structure that had the largest number of kinetochore proteins to date. We have removed the descriptions using the word "complete" from the text to avoid confusion. We note that the structures are not entirely consistent with some data, but the lack of a newer structure makes it hard for us to say they are not correct.

- Hinshaw and Harrison (2019) reported Ame1-Okp1 binding to Cse4 in a pulldown assay and showed how this likely fits into the overall kinetochore structure at the same time or before this was reported by the papers cited on this point, so this is an odd omission.

This was an oversight since we referenced the newer Hinshaw work and forgot to include the other older reference. We apologize for this and have added the appropriate 2019 reference to the revision.

- Phosphorylation-dependent inner kinetochore stabilization has recently been described in yeast (Klemm et al, 2023, Hinshaw et al, 2023, Hagemann et al, 2022) and vertebrates (Watanabe et al, 2019, Ariyoshi et al, 2021). CENP-C and Mif2 are Ipl1 targets, though the functions of these sites are not known. Can the authors rule out Mif2/CENP-C phosphorylation-dependent stabilization as the reason for the Ipl1-AID results?

We appreciate this point (as discussed above) and have mentioned in the Discussion that it is likely that additional Ipl1 substrates exist that contribute to Cse4 stability and kinetochore assembly.

- It would be great to know whether Cse4-END binds AO in the assembled kinetochore or if the mutations that disrupt this interaction just prevent something stable from assembling. Can this be inferred from the data somehow? It would help clarify whether

the absence of Cse4-AO contact in cryo-EM reconstructions means the biochemical reconstitutions are missing something.

This is a critical question that we also would love to answer. We do not currently have a way to address this, although we do have preliminary data suggesting there are more copies of OA than the rest of the CCAN in assembled kinetochore. This suggests that there are two pools of OA. However, without a more concrete demonstration that the OA bound to Cse4 is maintained in vivo, we did not speculate on this issue in the revision.

o "Although the NDT is essential, it is absent..." - Since Cse4-END is structured when bound to AO, it's not quite right to say that Cse4-N disorder is what prevents us from seeing it in the cryo-EM structures.

This is an excellent point and we have therefore removed the sentence.

- Some of the figure titles, especially in the supplement, seem to be incorrect. Titles for figures S1 and S2 should be updated.

We appreciate the reviewer noticing this and have ensured that all figure titles are correct.

- Figure S7B and other imaging experiments: were these cultures staged or asynchronous?

We have clarified the cell cycle stage for all imaging experiments.

- Page 14: cse4-S40,L41A the Cse4 mutation is incomplete

We appreciate the reviewer catching this mistake and have completed the nomenclature in the text.

- Page 16: "...so we presume [Scm3] initially targets Cse4 to centromeres via [Cse4 histone fold]..." but we don't know whether these interactions are mutually exclusive.

We also appreciate this suggestion and have modified the text accordingly.

Referee #3:

Title: Stable centromere association of the yeast histone variant Cse4 requires its essential N-terminal domain.

Authors: Popchock et al.

Popchock et al., describe a new molecular mechanism for Cse4 stabilization at

centromeres using elegant technique using single molecule fluorescence assays to monitor Cse4 during kinetochore assembly. These studies define a novel role for the essential N-terminus domain (END) of Cse4 in kinetochore assembly. More specifically they describe a role for phosphorylation of S40 in the END domain previously shown to be a substrate for Ipl1 by Boeckmann et al., 2013. They determined that Ipl1-mediated phosphorylation of Cse4-S40 plays a role in Scm3-mediated Cse4 stabilization at the kinetochore. Furthermore, the results show that interaction of essential components namey Okp1/Ame1 (OA) of the COMA complex and the interaction of Scm3 with END act independently to stabilize Cse4 at centromeric chromatin. A phosphomimetic mutant of Cse4 (Cse4 40SD) in the Cse4 END enhances Scm3 binding and can restore Cse4 recruitment in mutants defective in OA binding. The research objectives of this study are logical, and experimental approaches are robust and technically sound. The data presented in the manuscript are thorough, statistically methods used for data analysis are accurate, and data provides molecular insights into the role of Cse4-END in kinetochore function and chromosome segregation and should be of broad interest to the chromosome biology community. Listed below are comments.

Major Comments:

1. OA has not been defined in the abstract. Please do so [EDITOR's NOTE: please avoid the unnecessary abbreviation OA throughout the main manuscript text, and rather stick with Okp1/Ame1 for clarity]. Authors have shown in Figure 1B that depletion of OA affects the Cse4 nucleosome occupancy at the kinetochores. It is however unclear whether the phenotype observed here is unique to Cse4 or other inner kinetochore proteins are also affected. Given the extensive studies with human cells on the interaction of CENP-A and CENP-C (Mif2 homolog in budding yeast) it is of interest to examine the levels of at least one more inner kinetochore protein for example, Mif2 upon OA depletion, which will give additional support to their observation and provide a deeper mechanistic insight into the role of OA in kinetochore assembly.

We have changed OA to Okp1/Ame1 throughout the revised manuscript. We have performed fiber-seq on additional CCAN mutants and none of them have a strong effect on the Cse4 nucleosome. We have added the Ctf19 deletion data to the Expanded View Figure 1C to demonstrate that another CCAN mutant does not have an effect on Cse4 nucleosome occupancy. We will report the rest of the mutant data in our upcoming fiber-seq manuscript rather than put it into this paper since it is not related to the OA story.

2. Authors have conducted experiments to examine the role of END domain of Cse4 and defined its novel function in the kinetochore assembly regulated by the phosphorylation of Cse4-S40. However, Cse4 contains S33 in the END that is also phosphorylated (Boeckmann et al. 2013 MBoC). A role for S33 phosphorylation in centromeric deposition of Cse4 was reported by Hoffman et al., 2018. The authors should examine the role of S33 as well, double mutants and single mutants for both residues. What is the phenotype of S40A or S33A in their assays? For example, in Figure 7A, the effect of cse4 S40A or S33A on growth of L41A.

The Cse4 S33 phosphosite is a Cdc5 site so we did not initially focus on it in this manuscript. However, to address this point, we performed TIRFM endpoint assays to analyze Cse4 in a *cdc5* mutant that have been added to the revision (Appendix Figure S1). Because we did not detect any altered Cse4 localization in the absence of *cdc5*, we did not further pursue working on the S33 mutant.

3. How do they know S40D just causes a structural change rather than being phosphomimetic? Why not use S40E?

We do not know if the S40D mutation causes a structural change or is strictly a phosphomimetic. However, the same issue will be true for an S40E mutant.

4. Authors have concluded that Scm3 interacts with Cse4-CATD for the recruitment of Cse4 at the kinetochores. Once Cse4 nucleosome assembles on the CEN DNA, Scm3 dissociates from Cse4-CATD and then interacts with the Cse4-END. This implies that Scm3 and Cse4-END interactions most likely may be occurring outside of the S-phase. It is important to address the cell cycle dependent roles for the interaction of OA and Scm3 with END domain in stabilization of Cse4 at centromeric chromatin using extracts from synchronized cells. Along the same lines, when does S40 get phosphorylated by Ipl1 and whether this helps to address the differential roles of Ipl1-mediated phosphorylation in error correction during mitosis versus stabilization of Cse4? Extracts from synchronized cells should help them address possible differential role of Ipl1 mediated phosphorylation, Scm3, OA interact with END domain and provide a better explanation for their results.

The reviewer brings up very interesting questions but we do not have an easy way to address them. The precise timing of when Scm3 interacts with the Cse4 END domain is difficult to determine and we do not have a way to analyze the timing of Ipl1 phosphorylation of Cse4 S40 in vivo (we do not have a phospho-specific antibody). In addition, it isn't clear that the timing of Ipl1 regulating the END domain needs to be different from its separate role in error correction. Kinetochores assemble within minutes after DNA replication (doi: [10.1101/gad.449407](https://doi.org/10.1101/gad.449407)), so there is not an easy way for us to address this with the TIRFM assay. We tried to address this by assaying assembly in G1 and there was no stable Cse4 assembly so we don't have an easy way to determine if Scm3 interacts with Cse4 outside of S phase in this assay. Here is the data:

5. Some observations from previous studies cannot be explained by the current study. This presents a conundrum for this reviewer. For example, the N-terminal domain of Cse4 (lacking 129 amino acids) containing END is dispensable for haploid growth when overexpressed (Morey et al. 2004, Eukaryotic Cell). The authors have mentioned this data without further discussion of how their data can explain these observations. The cells are alive even without END... how does one explain the role of OA and Scm3 in these cells that lack the N-terminal domain? Can the C-terminus of Cse4 recruit Ame1 and Opk1. Along the same lines, overexpression of Cse4 can suppress *scm3* depleted cells. It is important for authors to include a discussion on these observations to reconcile their data.

We thank the reviewer for bringing up these points. Our assumption is that the overexpression of Cse4 likely contributes to its stabilization at the centromere so cells are able to live without the additional OA and SCm3 stabilizing mechanisms. We believe the events we have identified likely help with the initial assembly and then could become dispensable and Cse4 overexpression may therefore allow them to be bypassed by driving its own stability. We have added these points to the Discussion as requested (p. 17).

6. Figure 7 shows growth defects of *cse4*-mutants at high temperature and a correlation with the de-clustering of Cse4-GFP foci at the kinetochores. It is of interest to examine whether these mutants exhibit chromosome segregation defects?

To analyze chromosome segregation, the spindle checkpoint needs to be inhibited to allow cells to progress into anaphase. However, when we crossed the mutant to a checkpoint deletion, the cells were dead (Figure 7C). Because we also showed a clear issue with kinetochore-microtubule attachments (Figure 7D), we do not think trying to assay chromosome segregation will add new information and would require difficult strain construction due to the synthetic lethality with the checkpoint mutant.

7. Model (Fig 8) describes the dual function of Scm3 via Cse4 CATD and NTD domains. Is there any evidence that CATD-bound Scm3 is released when DNA wraps to form nucleosome? Is this a relocation?

Carl Wu's lab showed that Scm3 is released when Cse4/H4 interacts with DNA (Figure 4E, doi: 10.1016/j.molcel.2011.07.009). We therefore do propose that the Scm3 binding to the Cse4 N-terminus is a relocation and have added this point to the figure legend for the model Figure 8.

Other comments:

1. Why not also examine assembly of another COMA component, Ctf19 to the centromere in the context of END, and S40D. Unlike Ame1 and Okp1, Ctf19 is not essential, so extracts from a *ctf19*Δ strain will address the role of Ame1 and Okp1 is related to stabilizing Cse4 at centromere or some other role?

To address this question, we performed Cse4 TIRF endpoint assays in lysates lacking Ctf19. We did not detect a role for the Ctf19 protein in stabilizing Cse4 and have added this data to the manuscript (Appendix Figure 1A).

2. If phosphorylation of S40 is essential for stabilization of Cse4 at CEN, we may expect the Cse4 4SA will cause defects in chromosome segregation but based on Boekmann et al., 2013 the Cse4 4SA does not exhibit defects in chromosome segregation. How do the authors reconcile the role of phosphorylation based on these observations?

While the role of OA binding to the Cse4 END domain is essential, the mutant phenotypes and our data suggest that phosphorylation of Cse4 at S40 is a less important function. Indeed, the *cse4-S40A* mutant does not exhibit any major phenotypes and we believe this is just a contributor to Cse4 stability but not an essential function. This comment is similar to Reviewer 2's point that additional Ipl1 substrates may be involved in stabilizing Cse4 and we have added this point to the Discussion.

3. Lethality of Scm3 depletion can be rescued by OE Cse4, I am wondering if under this condition whether OA associate with the centromere? Will this phenotype be altered upon deletion of CAF1?

We agree these are interesting questions but are not central questions to our study. Testing this will require a lot of new strain construction and experiments that will not fit into a clear part of the results section of this manuscript.

4. Curious why authors do GST pull-down instead of in vivo CoIP?

We did a GST pulldown with recombinant proteins because it is a more rigorous way to demonstrate direct interactions. A co-IP from cells reports on both indirect and direct interactions so it is harder to make conclusions about whether the interactions are direct.

5. Previous studies have shown that Cdc5 (Mishra et al., 2019)) can also phosphorylate S40, have the authors examined the role of Cdc5 mediated phosphorylation of S40 in interaction with Scm3?

As mentioned above, we have now performed a Cse4 TIRF endpoint assay from a *cdc5* mutant lysate and found that Cdc5 does not regulate Cse4 stability. This is reported in Appendix Figure S1.

6. They have demonstrated a novel interaction of Cse4-END domain with its chaperone Scm3. Previous studies have shown that CATD of Cse4 interacts in vivo and in vitro with the HR-domain of Scm3 (Stoler et al. 2007 PNAS; Mizuguchi et al. 2007, Cell). It will be of great interest to define the domain within Scm3 that interacts with the Cse4-END domain.

We agree that it will be very interesting to learn more about the binding interface in the future. The Kumar lab (doi.org/10.1016/j.str.2024.03.002) initially identified the Scm3/END interaction and were not able to identify a separation of function mutant, likely because Scm3 is highly unstructured (as noted by reviewer 2).

Dr. Sue Biggins
Fred Hutchinson Cancer Center
Division of Basic Sciences
1100 Fairview Ave N
Mailstop A2-168, PO B0x 19024
Seattle, WA 98109

29th Nov 2024

Re: EMBOJ-2024-118553R

Stable centromere association of the yeast histone variant Cse4 requires its essential N-terminal domain

Dear Sue,

Thank you for submitting your revised manuscript to The EMBO Journal. Two of the original referees have now assessed it once again (see comments below), and both of them are overall satisfied with the revisions. Referee 2 retains a few minor concerns (see their attached report in Word format for clarity), which will require a few additional presentational changes. Once these have been incorporated in a final round of minor revision, we should be ready to proceed with formal acceptance and publication of the study.

In addition, please also take care of the following remaining editorial issues at this final stage:

- Please carefully check the reference list, since several citations (e.g. Akyoshi et al, Bohaczuk et al, Jha et al) appear to be incomplete. Also, for the preprint citation of Bohaczuk et al, please adjust the citation format as specified in our author guidelines for preprints:

The citation in the text should be: "(preprint: NAME1 et al, YEAR)"

The citation in the reference list: "Author NAME1, Author NAME2, ... (YEAR) article title. bioRxiv/ResearchSquare doi: XXX"

- Please check for congruency of the funding information listed in the manuscript and entered in our submission system; CA015704 is provided for NIH in the latter, while in the manuscript this number appears acknowledged for Fred Hutch/University of Washington Cancer Consortium.

- Please rename the Mat/Meth section into Methods, and upload the Reagents & Tools table separately.

- For Figure 4, please make sure to reference both subpanels A & B at least once in the text.

- As we are switching from a free-text author contribution statement towards a more formal statement based on Contributor Role Taxonomy (CRediT) terms, please remove the present Author Contribution section and instead specify each author's contribution(s) directly in the Author Information page of our submission system during upload of the final manuscript. See <https://casrai.org/credit/> for more information.

- Our routine pre-acceptance text similarity checks found several instances of Material & Methods section passages being near-verbatim copies from previous papers by yourself or your coworkers (Herman et al, Popchok et al). Although we encourage this in the interest of reproducibility, please however make sure to cite the earlier works on those occasions, by prefacing the respective sections with something like "xxx was done essentially as in yyy et al; briefly/specifically, ...".

- Finally, please provide suggestions for a short 'blurb' text prefacing and summing up the study in two sentences (max. 250 characters), followed by 3-5 one-sentence 'bullet points' with brief factual statements of key results of the paper; they will form the basis of an editor-written 'Synopsis' accompanying the online version of the article. Please also upload a synopsis image, which can be used as a "visual title" for the synopsis section of your paper. The image should be in PNG or JPG format with the modest dimensions of EXACTLY 550 pixels wide and 300-600 pixels high (this could be simply a slightly condensed version of Figure 8).

Please do not hesitate to contact me should you have any questions regarding these remaining presentational revisions. I look forward to receiving your final version!

With kind regards,

Hartmut

*** PLEASE NOTE: All revised manuscripts are subject to initial checks for completeness and adherence to our formatting guidelines. Revisions may be returned to the authors and delayed in their editorial re-evaluation if they fail to comply to the following requirements (see also our Guide to Authors for further information):

9) To facilitate reproducibility and cross-laboratory adoption of methodologies, please structure the Materials & Methods section as outlined in our guide to authors, including a completed Reagents and Tools Table that can be downloaded from our author guidelines as well (<https://www.embopress.org/page/journal/14602075/authorguide#structuredmethods>).

10) Digital image enhancement is acceptable practice, as long as it accurately represents the original data and conforms to community standards. If a figure has been subjected to significant electronic manipulation, this must be clearly noted in the figure legend and/or the 'Materials and Methods' section. The editors reserve the right to request original versions of figures and the original images that were used to assemble the figure. Finally, we generally encourage uploading of numerical as well as gel/blot image source data; for details see: embopress.org/page/journal/14602075/authorguide#sourcedata

At EMBO Press, we ask authors to provide source data for the main manuscript figures. Our source data coordinator will contact you to discuss which figure panels we would need source data for and will also provide you with helpful tips on how to upload and organize the files.

In the interest of ensuring the conceptual advance provided by the work, we recommend submitting a revision within 3 months

(27th Feb 2025). Please discuss the revision progress ahead of this time with the editor if you require more time to complete the revisions. Use the link below to submit your revision:

Link Not Available

Referee #2:

We are grateful to the reviewers and editor for their thoughtful suggestions on how to improve the manuscript. We have made the following revisions in response to the comments:

Referee #2:

The updated manuscript is improved, largely by clarification and omission. The sole resultant shortcoming, as far as I can tell, is that there is now no figure showing with purified proteins that Cse4-S40D alters Cse4-Scm3 binding. However, it is fine to leave this for another time unless the authors have developed an FP or similar assay that they are ready include in this manuscript. As before, I strongly support publication of the manuscript. This time, I have only minor concerns. I have entered these in blue text blow. In my view, the most important thing at this point is to provide some context for a reader who may not initially realize that the Cse4-END-Smc3 interaction is a strange one compared with conventional peptide binding interactions.

20240807

In this paper, Popchock et al. study protein interactions at the inner kinetochore. The major strength of this manuscript is that it uses two powerful techniques to bring totally new information to the table. The first technique, which is used only in the first figure, is a chromatin fiber mapping technology, which conclusively shows the DNA footprint of the centromeric nucleosome/CCAN. This is exciting to see. The second technique, which is used throughout the manuscript, merges an optimized in vitro kinetochore reconstitution strategy with TIRF microscopy, enabling the authors to visualize and measure individual kinetochores. The same authors reported this technique in a separate recent publication, so the assay itself is not totally new, but the sky is the limit for what this experiment and its variants can tell the field. More, please!
The main conclusion of the paper is that both Okp1-Ame1 and Scm3 stabilize the centromeric histone protein, Cse4, at centromeres. Ctf19 complex (Ame1-Okp1)-dependent Cse4 stabilization has been shown in meiotic prophase and is less surprising. Scm3-dependent Cse4 stabilization even after Cse4 deposition is somewhat surprising. The major strength of the paper is the ability to directly visualize and quantify these binding interactions in a near-native context, which is totally amazing. The major weakness of this paper is that the nature of the interaction between Scm3 and the Cse4 N-terminal region is not well-described. Previous work on this subject has not been definitive, and the biochemical assays done in the current manuscript are suggestive but leave substantial room for argument, especially regarding Cse4 phosphorylation and its function. There are clear genetic and functional data showing that Cse4-S40D can indeed rescue cells from inefficient Ame1-Okp1-Cse4 contact, which is new and interesting.

My recommendation is that the authors rework the paper to remove most of the biochemical Scm3-Cse4 binding data/claims and focus instead on the phenomena that are clearly supported by the data: Ame1 and Scm3 both are required for stable Cse4-CEN association. Cse4-S40D can suppress Ame1-Okp1 binding defects. Ipl1 enhances Cse4 loading.

Below, I pose specific questions. I do not think the authors need to include quantitative biochemical assays to publish this work, but I think they essentially have a choice between expanding the binding assays to make them quantitative or removing several of the current pulldown-style experiments and publishing a paper that focuses much more on the phenomenon, substantiated by genetic data, rather than a direct binding interaction. I am strongly in favor of publishing this paper after addressing these concerns.

Major questions:

- The idea that Cse4 phosphorylation controls differential affinity for Scm3 and Ame1/Okp1 is plausible, but the current data is not convincing on this point.
 - o The pulldown data showing differential binding of phospho-mimetic Cse4-N (S40D) to Scm3 is not rigorous, since the different ratios used are not meaningful in a biochemical sense.
 - o To determine whether S40D enhances binding affinity, a quantitative assay like fluorescence polarization or ITC should be done (see comments above).

To address the concerns of this reviewer, we have moved some of the biochemistry data to the appendix and somewhat downplayed the conclusion that the phosphorylation directly controls Scm3 binding throughout the paper. We have tried to focus on Ipl1 phosphorylation of the END domain instead where possible instead of saying Scm3 binding. However, the TIRF data does show an enhanced stability of Scm3 at centromeres when Cse4 is phosphorylated on S40, so there is some clear effect of this phosphorylation event on Scm3 localization at the centromere through the Cse4 N-terminus. In addition, we have also recapitulated the published data showing the binding of Scm3 to the END domain, so we think it is reasonable to state that

Scm3 binds to the END domain.

I agree. It is reasonable and has been shown clearly.

- The cited published data for an Scm3-Cse4-N interaction is not definitive. The authors of the cited paper suggest the following: "We have established that both proteins are intrinsically disordered and form a protein complex in vitro with no significant gain in structure." Is this the view of the authors of the current paper? I'm struggling to understand how I should think about the proposed Scm3-Cse4-N interaction. Adding to the confusion, the current manuscript shows that Cse4-N delta-END does not bind Scm3 in a pulldown assay. Since Cse4-END is helical when bound to Ame1-Okp1 and in all predictions, this seems at odds with the published Scm3-Cse4 binding data.

o The cited binding data also suggests Scm3 binds AO. Do the authors think this is relevant? And has this been tested?

o Alpha fold 3 predictions show no meaningful interaction between Scm3 and Cse4-N (unmodified or phospho-S22, -S30, -S40). This is very different from the AO-Cse4-N predictions.

We did not mean to imply that Scm3 binding to the END domain would have the same effect as OA binding to this domain. While we understand that this reviewer is skeptical about the interaction, we have confirmed the published NMR data (doi.org/10.1016/j.str.2024.03.002) with biochemical pulldown assays so we don't think it should be entirely ignored despite alpha fold 3 not predicting how they interact. As brought up in other comments, it is entirely possible Scm3 has many additional interactions with other kinetochore proteins and the "fuzzy" interaction between Cse4 and Scm3 is just one of many. We have therefore decided to leave some of the Scm3 binding in the manuscript but have downplayed its importance. We agree there are structural questions about this interaction that need to be tackled in the future.

I would ask the authors to consider more precisely what is meant by a "fuzzy" interaction. If they are held together by relatively unspecific charge-charge interactions, wouldn't solvation be sufficient to break apart Cse4-END and Scm3 under physiologic conditions or at the very least in high salt? Does the pulldown only work because of high avidity on the beads? I do not agree that this is a purely "structural" question that can be so easily cleaved from observation of the interaction by pulldown, and so the reader should be given a heads up and some help thinking about this. Thus, I do think the manuscript needs to acknowledge the interaction as presented invokes binding of two peptides with no secondary structure when associated, or (genuinely) is there another possibility here? Can the authors reference another such interaction that has been extensively validated? My use of AF3 was simply a way of illustrating the oddity of the situation.

- The diagram at the end is not fully supported by the data (see above). Here are some specific questions: i) why isn't CBF3/Ndc10 preloaded on the CEN DNA? ii) Why does Scm3 appear to fall off in steps 4 and 5? iii) Isn't Scm3 recruited through CBF3? Is there a reason to think this interaction would not persist (allowing for exchange, which is known)? In my view, a more conceptualized schematic showing mainly the studied factors (CEN, Scm3, Cse4, and AO) and their observed relationships/dependencies in the reconstitution assay would be more helpful here. I don't think the protein-suggestive cartoons are necessary, since they give the impression of a worked out structural mechanism, but that is not the case.

We appreciate the suggestion to focus the model and did not mean to imply more than we discovered. We have simplified the model (Figure 8) in the revision to focus only on the observations made in the manuscript and therefore left out other factors such as CBF3.

- The S40 site is reported as an Ipl1 site, but the published data supporting this idea is not conclusive. Are there examples of known R/K - X - X - S/T - L/I Ipl1 sites? The data showing Ipl1-dependent Cse4 stabilization and compensation by S40D hold together, but the site itself and the relatively weak underlying data is an uncomfortable fact. Are there other potential substrates that may explain the Ipl1-AID and S40D rescue results? MIND, which might stabilize everything, is probably not recruited at high enough efficiency (fractional occupancy). What about Mif2? Ndc10? Both are Ipl1 substrates, of course.

It was previously published that Cse4-S40 is an Ipl1 site ([doi: 10.1091/mbc.E12-12-0893](https://doi.org/10.1091/mbc.E12-12-0893)). While S40 does not meet the common consensus site criteria, there are many examples of Ipl1 sites that have a slightly different consensus (for example, Mad3 Ser10: YSKKRISYMP ([doi: 10.1101/gad.431507](https://doi.org/10.1101/gad.431507)), Mif2 Ser277: RTDSIIDR ([DOI: 10.1016/j.cub.2023.01.012](https://doi.org/10.1016/j.cub.2023.01.012))). However, we agree that there are many additional Ipl1 targets at the kinetochore, including Mif2 and Ndc10, and we are interested in exploring their roles in Cse4 stability in the future. We have therefore made it clear in the revised manuscript that there may be one or more additional Ipl1 substrates involved in Cse4 stability and kinetochore assembly (Discussion, p. 18).

These references are helpful. It might be good to include them in a final version of the manuscript to buttress this point and help the reader interpret the claim/lay of the land.

A final manuscript may benefit from more careful evaluation by the authors of spots in the text where the experiments are said to examine Cse4 phosphorylation when instead a phosphomimetic mutation is used. It is admittedly cumbersome to be perfectly precise each time, and so it is probably not practical to do so, but the reader needs to be reminded at least once that the current manuscript neither shows direct evidence for Cse4 phosphorylation, nor does it directly evaluate this mark except possibly in the Ipl1-AID experiments (indeed, "direct evaluation" is an exceedingly high bar for any individual phosphosite).

- The authors propose that Ipl1 proximity serves a licensing function for Cse4 nucleosome assembly. It isn't clear why kinase activity should be required. Wouldn't CBF3 binding (for instance) be sufficient? Kinase activity might provide a temporal cue, but there is no obvious reason to need this, since the main assembly event in yeast is thought to be immediately after passage of the replication fork.

We agree that Ipl1 is not a master switch for Cse4 assembly. We propose it helps to license because it would be yet another layer of control at the kinetochore to stabilize the nucleosome at the proper site. CBF3 is indeed a much stronger control over Cse4 deposition but the stability seems to be controlled by other kinetochore proteins, such as Okp1, Ame1 and Scm3. We propose that these many controls are part of a licensing mechanism to stabilize it at the centromere and adding phosphoregulation as another regulatory event helps to ensure stability at the proper locus. We did not mean to imply it would be strictly required.

- Given their hypothesis about Cse4 licensing, do the authors think that the C0N3 loading is poor because the DNA gets loaded with normal (dark) H3 nucleosomes?

This is a really interesting question. In our prior work (DOI: 10.1101/2023.01.20.524981), we asked whether H3 assembled onto a very similar template and did not detect significant H3 assembly. Because this template had the Widom DNA that is in the C0N3 template, we don't believe the C0N3 template would be different and we have added this point to the text (p. 7).

Indeed, C0N3 is likely a much better substrate for H3 assembly than CEN3. It sounds like CDEI/III (and bound proteins) actively oppose nucleosome loading of any kind. The Fiber-Seq experiments are a great way to look at this.

Minor questions:

- There seems to be an extra protected region to the right of CEN5 in Figure 1B. Is this the R-CDEIII assembly site for a second CBF3? Has this been observed before? See PMID30478265, Figure 4.

There is indeed ~30bp of protected DNA downstream of the CDEIII element. While we don't know the nature of the proteins protecting this region, it agrees with recent structural data (DOI: 10.1126/sciadv.adg748) showing that CCAN extends ~30bp beyond CDEIII. However, as pointed out, it could potentially be a second CBF3 binding site. This extra protection has been observed before (<https://doi.org/10.1073/pnas.1104978108>). We will discuss this in our upcoming manuscript where we publish the full genomic analysis of the fiber-seq data. We did not include it here since it didn't seem relevant to the current manuscript focus but appreciate the reviewer noticing this interesting point.

- The image segmentation in figure 2C is questionable. Was the segmentation threshold manually defined? Could this be improved?

We did not realize this image appeared concerning. It was not handled differently from any other images but we replaced it to avoid concerns.

- The Cse4 internal GFP tag is not mentioned in the main text. I still see some papers using C-terminal Cse4 tags, and it would be good to emphasize the importance of the internal tag in a single short sentence/phrase somewhere.

We appreciate this suggestion and have made it clear we are using an internal tag (p. 7).

- The blot in Figure 4A looks like it was cut at an unlucky spot. Can this be rerun? Ideally, this and the blot in figure 4C would be done on the same membrane and in the same experiment so that the cse4-S40D mutant can be directly compared with WT.

Since these blots are not quantitative to begin with, we moved them to the appendix (Appendix Figure S1B and S1C) and left the quantitative TIRF results in the main manuscript.

- The experiment showing co-binding to Cse4-N by AO and Scm3 is not rigorous. There seems to be excess Cse4-N in this experiment, so it's not clear we should expect any competition in the first place, even with perfectly competitive binders.

We have removed this experiment given the reviewer's concern since it was not important to the conclusions of our manuscript.

- It looks like the Cse4 off rates do not match Popchock, 2023. For example, Figure 2D in 2023 vs. Figure 3E in the current manuscript. Can the authors explain this? If these are different lysates, is there a good way to quantify the lysate-specific activity? What parameters vary between lysates (off rate? Final loading amount? Etc.) and does this correlate with concentration, cell health, etc?

The reported off rates are not analyzing Cse4 alone so they cannot be compared. Instead, the off-rate analysis is measuring a ternary complex of two proteins with the centromeric DNA. In this manuscript, we analyzed the off rate of Cse4 with either Scm3 or Ame1. In the previous paper, we analyzed Cse4 with Ndc10 (Figure 3E, 2023).

- Figure 6B and C look like they use the same AME1 WT data. Shouldn't these just be shown in a single plot? In other words, two plots for the figure, one for Cse4 and one for Ame1 localization? Looking more closely, this also seems true of Figure 4B-D (IPL1, and ipl1-AID). Why not just merge them?

We merged the graphs in Figure 4B as requested. In Figure 6, there are two different proteins being assayed (Cse4 and Ame1) so we need to keep separate graphs for each. In addition, there are various mutants for each protein, so we kept these graphs separate for clarity.

I am confused about Figure 6 - it seems like B and D could be merged/reorganized so that Cse4 endpoint data is in one graph (all genotypes) and Ame1 data likewise? Perhaps I am missing a key point? If the current organization is done just to keep groups of genotypes separate, I suppose this is okay if judged essential, but then the legend should be straightforward about saying the data is repeated for WT.

- In many places, the authors reference structures of the assembled kinetochore, but (as it is rightly acknowledged, though not firmly enough) the referenced cryo-EM structures probably do not reflect the physiologic assemblies, so it is strange to reference them on the one hand as models for what is assembling in extract while on the other hand present and point to previous data (Popchok, 2023, for instance, and in the cited papers themselves) that indicates these nucleosome-containing structures are incorrect.

o "...sequence used in the most recent and complete structural studies..." is a great example. What is meant by "complete" here? There seems to be a logical disconnect.

Our use of the word "complete" referred to the structure that had the largest number of kinetochore proteins to date. We have removed the descriptions using the word "complete" from the text to avoid confusion. We note that the structures are not entirely consistent with some data, but the lack of a newer structure makes it hard for us to say they are not correct.

- Hinshaw and Harrison (2019) reported Ame1-Okp1 binding to Cse4 in a pulldown assay and showed how this likely fits into the overall kinetochore structure at the same time or before this was reported by the papers cited on this point, so this is an odd omission.

This was an oversight since we referenced the newer Hinshaw work and forgot to include the other older reference. We apologize for this and have added the appropriate 2019 reference to the revision.

- Phosphorylation-dependent inner kinetochore stabilization has recently been described in yeast (Klemm et al, 2023, Hinshaw et al, 2023, Hagemann et al, 2022) and vertebrates (Watanabe et al, 2019, Ariyoshi et al, 2021). CENP-C and Mif2 are Ipl1 targets, though the functions of these sites are not known. Can the authors rule out Mif2/CENP-C phosphorylation-dependent stabilization as the reason for the Ipl1-AID results?

We appreciate this point (as discussed above) and have mentioned in the Discussion that it is likely that additional Ipl1 substrates exist that contribute to Cse4 stability and kinetochore assembly.

- It would be great to know whether Cse4-END binds AO in the assembled kinetochore or if the mutations that disrupt this interaction just prevent something stable from assembling. Can this be inferred from the data somehow? It would help clarify whether the absence of Cse4-AO contact in cryo-EM reconstructions means the biochemical reconstitutions are missing something.

This is a critical question that we also would love to answer. We do not currently have a way to address this, although we do have preliminary data suggesting there are more copies of OA than the rest of the CCAN in assembled kinetochore. This suggests that there are two pools of OA. However, without a more concrete demonstration that the OA bound to Cse4 is maintained in vivo, we did not speculate on this issue in the revision.

o "Although the NDT is essential, it is absent..." - Since Cse4-END is structured when bound to AO, it's not quite right to say that Cse4-N disorder is what prevents us from seeing it in the cryo-EM structures.

This is an excellent point and we have therefore removed the sentence.

- Some of the figure titles, especially in the supplement, seem to be incorrect. Titles for figures S1 and S2 should be updated.

We appreciate the reviewer noticing this and have ensured that all figure titles are correct.

- Figure S7B and other imaging experiments: were these cultures staged or asynchronous?

We have clarified the cell cycle stage for all imaging experiments.

- Page 14: cse4-S40,L41A the Cse4 mutation is incomplete

We appreciate the reviewer catching this mistake and have completed the nomenclature in the text.

- Page 16: "...so we presume [Scm3] initially targets Cse4 to centromeres via [Cse4 histone fold]..." but we don't know whether these interactions are mutually exclusive.

We also appreciate this suggestion and have modified the text accordingly.

Referee #3:

Studies by Popchock et al., describe a new molecular mechanism for Cse4 stabilization at centromeres using single molecule fluorescence assays to monitor Cse4 during kinetochore assembly. These studies define a novel role for the essential N-terminus domain (END) of Cse4 and show that phosphorylation of S40 in the END domain by Ipl1 plays a role in Scm3-mediated Cse4 stabilization at the kinetochore. Importantly the essential components namely Okp1/Ame1 (OA) of the COMA complex and the interaction of Scm3 with END act independently to stabilize Cse4 at centromeric chromatin.

The authors have addressed the comments and for several of these they provided an explanation for why some of experiments are not feasible or beyond the scope of this work. Data for two additional experiments 1) *ctf19* deletion (Expanded view Fig 1C) and 2) *cdc5* mutant (Appendix Figure S1) have been added. The authors mention that results with CENP-C/Mif2 will be presented in another paper. Several other comments were addressed by revision to the text.

The research is well designed and the data provides molecular insights into the role of Cse4-END in kinetochore function and chromosome segregation which should be of broad interest to the chromosome biology community.

All editorial and formatting issues were resolved by the authors.

Dr. Sue Biggins
Fred Hutchinson Cancer Center
Division of Basic Sciences
1100 Fairview Ave N
Mailstop A2-168, PO B0x 19024
Seattle, WA 98109

6th Dec 2024

Re: EMBOJ-2024-118553R1
Stable centromere association of the yeast histone variant Cse4 requires its essential N-terminal domain

Dear Sue,

Thank you for submitting your final revised manuscript for our consideration. I am pleased to inform you that we have now accepted it for publication in The EMBO Journal.

With kind regards,

Hartmut
